

# Impact of land-water sensitivity contrast on MOPITT retrievals and trends over a coastal city

Ian Ashpole[1] and Aldona Wiacek[1,2]

[1]Department of Environmental Science, Saint Mary's University, Halifax, Canada
[2]Department of Astronomy and Physics, Saint Mary's University, Halifax, Canada

*Correspondence to:* Ian Ashpole (ian.ashpole@smu.ca)

**Abstract.**

We compare MOPITT Version 7 (V7) Level 2 (L2) & Level 3 (L3) carbon monoxide (CO) products for the 1º x 1º L3 gridbox containing the coastal city of Halifax, Canada, for the seasons DJF and JJA, and highlight
a limitation in the L3 products that has significant consequences for the temporal trends in near-surface CO identified using those data. Because this gridbox straddles the coastline, the MOPITT L3 products are created from the finer spatial resolution L2 products that are retrieved over both land and water, with a greater contribution from retrievals over water because more of the gridbox lies over water than land. We create alternative L3 products for this gridbox by separately averaging the bounded L2 retrievals over land ($L3_L$)
and water ($L3_W$) and demonstrate that profile and total column CO (TCO) concentrations, retrieved at the same time, differ depending on whether the retrieval took place over land or water. These differences ($\Delta$RET) are greatest, and most significant, in the lower troposphere (LT), with maximum mean differences of 11.4 % (14.9 ppbv, p = 0.116) at the 900 hPa level in DJF, and 10.8 % (12.4 ppbv, p = 0.005) at the surface profile level in JJA. Retrieved CO concentrations are more similar, on average, in the middle and upper troposphere
(MT and UT), although large differences (in excess of 50 %) do infrequently occur. Significant (p < 0.1) TCO differences of ~5 % are also found in both seasons. By analyzing $L3_L$ and $L3_W$ retrieval averaging kernels and simulations of these retrievals, we demonstrate that, in JJA, $\Delta$RET is strongly influenced by differences in retrieval sensitivity over land and water, especially close to the surface where $L3_L$ has significantly greater information content than $L3_W$. In DJF, land-water differences in retrieval sensitivity are
much less pronounced and appear to have less of an impact on $\Delta$RET, which analysis of wind directions suggests is more likely to reflect differences in true profile concentrations (i.e. "real" differences). The original L3 timeseries for the 1º x 1º gridbox containing Halifax ($L3_O$) corresponds much more closely to $L3_W$ than $L3_L$, owing to the greater contribution from L2 retrievals over water than land. Thus, in JJA, variability in retrieved CO concentrations close to the surface in $L3_O$ is suppressed compared to $L3_L$, and
they decline at a significantly slower rate (surface profile level trends of -1.16 (±0.32) ppbv y$^{-1}$ vs -3.28 (±0.68) ppbv y$^{-1}$, from $L3_O$ and $L3_L$ respectively). This is because contributing L2 retrievals over water are closely tied to a priori CO concentrations used in the retrieval, owing to their lack of near-surface sensitivity



in JJA, and these are based on monthly climatological CO profiles from a chemical transport model and therefore have no yearly change (surface profile level trend = -0.51 (±0.28) ppbv $y^{-1}$ in $L3_W$). Although we focus only on the city of Halifax, our results imply potentially large differences in the results of near-surface CO analysis using the L2 and L3 datasets for other cities that are situated within a coastal L3 gridbox, among which are some of the most populous in the world.

# 1 Introduction

The Measurement of Pollution in the Troposphere (MOPITT – Drummond et al., 2010, 2016; Acronyms defined in Appendix A) instrument is one of a large fleet of satellite-borne instruments capable of observing the composition of the Earth's atmosphere from space. The primary target gas for MOPITT is carbon monoxide (CO). Emitted from a range of anthropogenic (e.g. fossil fuel use) and natural (e.g. wildfires) sources, and with an atmospheric lifetime of weeks to months depending on season and location, CO is vital to monitor as a pollutant in its own right, as a tracer of local and transported pollution sources, and also because it plays an important role in atmospheric chemistry, i.e. as a precursor to ozone formation and a primary sink for the hydroxyl radical. While multiple sensors observe CO (see e.g. Worden et al. (2013) for a comparison of CO trends from four satellite instruments), the unique strength of MOPITT lies in its nearly unbroken record of observations since launch in 1999. This makes MOPITT data very valuable for the analysis of temporal trends in CO concentrations (e.g. He et al., 2013; Worden et al., 2013; Strode et al., 2016).

MOPITT retrieves coarse vertical resolution CO profiles in the troposphere by inverting observed upwelling radiances at thermal infrared (TIR) and near infrared (NIR) wavelengths (Deeter et al., 2013). These profiles are integrated to give CO total column amounts (TCO). In addition to several other inputs, MOPITT's optimal estimation retrieval algorithm requires a priori information – among which is a description of the most probable state of the CO profile and its variability – to obtain physically realistic results (Pan et al., 1998; Rodgers, 2000; the retrieval algorithm is outlined in more detail in Section 2.1.1). The proportion of information about CO concentrations in each individual retrieval that comes directly from the satellite measurement, as opposed to the a priori, is highly variable. It depends on scene-specific factors such as surface temperature, thermal contrast in the lower troposphere, and the actual ("true") CO loading itself, as well as on instrumental noise (e.g. Deeter et al., 2015). This complicates the interpretation of retrievals, thus placing great importance on the analysis of retrieval averaging kernels (AKs), which represent the sensitivity of each retrieved profile point to the true CO profile and quantify the overall information content of the retrieval (as described in detail by, e.g., Deeter et al. (2007, 2015) and Rodgers (2000)). The lower the retrieval information content, the closer the retrieved CO loading will be to the a priori, which is



based on a climatological model value. Retrievals with little information content should thus be treated with caution in any analysis.

In general, the greatest information content is associated with daytime retrievals over land, during the summer season (MOPITT Algorithm Development Team, 2017; Deeter et al., 2015). This is where and when thermal contrast conditions are typically greatest, maximizing the instrument's ability to sense CO absorption

in the lowermost layers of the troposphere against the hot surface emission background (Deeter et al., 2007; Worden et al., 2010). To ensure that analyses involving MOPITT data are not biased by retrievals that have a heavy reliance on the a priori (in other words, a low information content), it is therefore suggested that users of MOPITT data consider excluding from analysis retrievals obtained during winter months, over water, and also from certain other geographical areas where retrieval information content is known to be low, i.e.

over mountainous regions, where the effects of geophysical noise reduce information content relative to flatter terrain (MOPITT Algorithm Development Team, 2017; Deeter et al., 2015). Deeter et al. (2015) specifically emphasizes such filtering in the analysis of long-term CO trends, since inclusion of retrievals with a heavy a priori weighting will weaken any real trends in the data. This occurs because the a priori CO is based on monthly climatologies of modelled CO amounts and is therefore variable by month but not by

year (Deeter et al., 2014).

MOPITT data are available as Level 2 (L2) products, where each individual retrieval at 22 x 22 km spatial resolution is available for analysis; and Level 3 (L3) products, which are a 1° x 1° area-averaged version of the individual L2 retrievals that fall within each gridbox (with some filtering criteria applied – see Section 2.1.2). At the heart of this study is the fact that some L3 gridboxes straddle the coastline. L3 products for

such gridboxes can therefore be based on L2 retrievals over both land and water (see Figure 1), the information content of which can differ greatly (e.g., Deeter et al., 2007). In this study, we demonstrate, for a coastal L3 gridbox, how well known and well characterized differences in retrieval sensitivity over land and water can lead to significant differences in the L2 retrieved profiles that are averaged together to create the L3 products (Section 3.1). We outline the impact that this has on the statistics of the resulting L3 CO

profiles, and demonstrate the consequences that it has for temporal trend analysis with the L3 dataset, when compared to the results of the same analysis applied to the underlying L2 data that can be filtered by surface type to maximize information content (Section 3.2). This is an important issue to be aware of for two reasons: firstly, owing to their smaller file size, L3 data are better suited to long timeseries analysis than L2 data (~25 MB vs ~450 MB respectively, for a single daily, global file). Working with L3 data requires fewer computing

resources and, arguably, less technical expertise, making the MOPITT data more readily accessible to a greater number of users who are potentially less well-positioned to scrutinize the data. Secondly, 6 of the top 10 and 43 of the top 100 largest agglomerations by population in the world (population data taken from



www.citypopulation.de, valid at time of writing) lie within a coastal L3 gridbox, and it is such cities that are likely to be targets for analyses of temporal trends in air quality indicators. The results that we report here 100 are therefore relevant to each of these cities.

This paper is structured as follows: In section 2 we outline the data and methods used in this study, giving an overview of the MOPITT instrument, retrieval, and surface type classification that is relevant to our work. In Section 3 our results are presented and discussed, and conclusions are drawn in Section 4.

## 2. Data and Methods

### 2.1 MOPITT

### 2.1.1 Instrument and retrieval overview

MOPITT has been making routine observations almost continuously since March 2000. It is carried on board the polar-orbiting NASA Terra satellite, with a nominal altitude of ~705 km and an equatorial overpass time of ~10:30 and ~22:30 local time. The instrument is a nadir-viewing gas correlation radiometer, with a ground 110 resolution of 22 x 22 km. It observes radiances in two CO-sensitive spectral bands: the TIR at 4.7 μm, and the NIR at 2.3 μm. The TIR band is sensitive to both absorption and emission by CO and yields information on its vertical distribution in the troposphere (Pan et al., 1995, 1998). The NIR band measures reflected solar radiation, which constrains the CO total column amount and yields information on CO concentrations in the lower troposphere (LT), to which TIR radiances are typically less sensitive (Pan et al., 1995, 1998). Our 115 results are based on analysis of the TIR-NIR combined product, owing to its greater sensitivity to LT CO compared to the TIR- and NIR-only products which are also available (e.g. Deeter et al., 2017). We restrict our analysis to daytime-only retrievals (more information on data selection in Section 2.1.3).

Multiple other sources describe MOPITT's CO retrieval algorithm in detail (e.g., Deeter et al., 2003; Francis et al., 2017). Briefly, it employs optimal estimation (Pan et al., 1998; Rogers, 2000) and a fast 120 radiative transfer model (Edwards et al., 1999) to invert radiance measurements performed by the instrument to obtain CO concentrations. Additional inputs required include meteorological data (profiles of temperature and water vapour), surface temperature and emissivity, and satellite viewing geometry for the radiative transfer model; and a priori CO profiles, to constrain the inversion to physically reasonable limits. For latest MOPITT product versions, meteorological fields are extracted from the NASA Modern-Era Retrospective 125 Analysis for Research and Applications Version 2 (MERRA-2) reanalysis product; and a priori CO profiles are derived from a monthly CO climatology for the years 2000–2009, simulated with the Community Atmosphere Model with Chemistry (CAM-chem) chemical transport model at a spatial resolution of 1º x 1º (Lamarque et al., 2012) and then spatially and temporally interpolated to the time and location of the MOPITT





observation. As it is a multi-year climatology, the a priori features no yearly trend, i.e. values for a given
location and day of the year are the same every year. Surface temperature and emissivity values are retrieved
from the radiance measurements. Retrievals are only performed for cloud-free scenes, with cloud
screening based on collocated Moderate Resolution Imaging Spectroradiometer (MODIS) observations and
MOPITT's own radiances. CO profiles are retrieved on 10 vertical levels, with 9 equally spaced pressure
levels from 900 to 100 hPa and a floating surface pressure level. Reported values represent the mean CO
volume mixing ratio (VMR) in the layer immediately above that level. Retrievals are initially performed on
a $\log_{10}(VMR)$ scale, owing to large CO variability in the atmosphere.

Averaging kernels are produced for each retrieval and distributed with the data. The AK matrix (A)
quantifies the the sensitivity of the retrieved vertical profile to the "true" vertical profile, and depends on the
radiance weighting functions, instrument error covariance matrix, and a priori covariance matrix. Its
relationship to the retrieved profile ($X_{rtv}$), the 'true' profile ($X_{true}$), and the a priori profile ($X_{apr}$) is as follows:

$$X_{rtv} = X_{apr} + A(X_{true} − X_{apr}) \tag{1}$$

Thorough analysis of AKs is essential for understanding the physical significance of MOPITT's CO
retrievals. We discuss AKs in more detail in section 3.1.2.

The MOPITT retrieval algorithm is subject to continuous development, in line with improvements in
understanding of the changing instrumental characteristics and geophysical factors that affect the retrieval
sensitivity, and with periodic updates to the radiative transfer model (Worden et al., 2014). This prompts the
release of new product versions, with enhanced validation statistics against in situ CO observations. The
work presented in this manuscript is based on MOPITT Version 7 (V7) products (Deeter et al., 2017). We
analyse both L2 and L3 products (as outlined below).

**2.1.2. Surface type classification**

Both L2 and L3 data files come with a range of diagnostic fields and values, in addition to the averaging
kernel matrix, that can be used for filtering and interpreting retrievals. Of particular importance is the surface
index flag. Because retrieval information content is variable depending on surface type (Deeter et al., 2007),
each L2 retrieval is tagged according to whether it was performed over land, water, or a combination of the
two ("mixed") in the case of 22 x 22 km L2 footprints that straddle the coastline or contain significant bodies
of water. The surface index of each L3 gridbox is then based on the L2 retrievals that fall within the relevant
1° x 1° grid boundaries (Figure 1). Where more than 75 % of the bounded L2 retrievals have the same surface
index, only those retrievals are used to produce the L3 gridded value (the other L2 retrievals are discarded)
and the L3 surface index is set to that surface type. Otherwise, all L2 retrievals available in the L3 gridbox



are averaged together and the L3 surface index is set to "mixed" (this information is taken from the MOPITT Version 6 L3 data quality summary[1] – at the time of writing, no V7 L3 data quality summary was available).

The averaging together of retrievals with significantly different sensitivity profiles – as could be the case when averaging retrievals over land and water – is generally advised against, as this serves to dilute the information coming from the MOPITT observed radiances with information coming from the a priori, thus

increasing the dependence of the resulting CO profile values on the a priori profile. In fact, guidelines to maximize the information content of MOPITT data and minimize the influence of the a priori are to restrict analysis to daytime observations over land during the summer season, since this is when thermal contrast conditions are greatest, thus maximizing the instrument's ability to sense CO in the lowermost layers of the troposphere (MOPITT Algorithm Development Team, 2017; Deeter et al., 2015; Deeter et al., 2007).

**2.1.3. Study area, time period, and MOPITT data processing in this study**

Our analysis is based on MOPITT retrievals over the city of Halifax in Nova Scotia, Canada (Figure 1: longitude = -63.583, latitude = 44.651). Situated on the Atlantic coastline, Halifax is the major economic center in Atlantic Canada, with a population of 316,701 in the urban core of Halifax Harbour (from 2016 census statistics). The pollution environment of Halifax, which is an intermediate port city, was characterized

by Wiacek et al., (2018); it showed no exceedances of regulated gaseous contaminants, but nevertheless a substantial contribution of shipping emissions that is comparable to or greater than emissions from the city's vehicle fleet. All available MOPITT V7 L2 and L3 TIR-NIR files ("MOP02J" and "MOP03J" files, respectively) were downloaded from the NASA Earthdata portal (https://search.earthdata.nasa.gov). This covers the period 2000-03-03 to 2017-03-05. At the time of writing, more recent data are flagged as "beta"

files, which await a future retrospective processing after the annual hot calibration becomes available, and their use in scientific analyses is discouraged (Deeter et al., 2017). For brevity, we restrict our analysis to the winter (DJF) and summer (JJA) seasons, since these best encapsulate the different thermal contrast conditions over land and water, when compared to the intermediate (MAM and SON) seasons.

We extract L3 data for the 1° x 1° gridbox that contains the city of Halifax, and retain only the observations

that were made during daytime hours. This yields a timeseries with one observation per day, when retrieval data were available within this gridbox (there are no retrievals available on 93 % of all days in DJF and 83 % of all days in JJA for the period covered – this is a result of 1) MOPITT's polar orbit limiting temporal resolution to ~3 days over most of the globe; and 2) retrievals either not being made due to cloud coverage, or discarded due to data quality issues, on days when the satellite swath encompasses Halifax). For clarity,

---

[1] available here: https://eosweb.larc.nasa.gov/sites/default/files/project/mopitt/quality_summaries/mopitt_level3_ver6.pdf



we refer to this original, "as-downloaded" L3 timeseries as $L3_O$ for the remainder of this paper, owing to the way that we process the L2 data (explained below). Because this gridbox straddles the coastline, the $L3_O$ surface index varies each day. The surface classification breakdown of the $L3_O$ timeseries is given in Table 1a. "Water" is the modal classification in both seasons, followed by "mixed". $L3_O$ is only classified as "land" on one occasion each season. This is most likely due to the fact that more of the L3 gridbox is situated over

water than land (Figure 1). The ratio of water to mixed observations is far greater in DJF than in JJA. This may be due to preferential cloud coverage over land in winter, and/or could be linked to the misidentification of snow/ice coverage on the surface as cloud during cloud screening (identifying the exact cause for this difference is beyond the scope of this paper).

We select all L2 retrievals that fall within the $1^o$ x $1^o$ L3 gridbox that contains the city of Halifax (lower-

left corner: $-64^o$ E, $44^o$ N; upper right corner: $-63^o$ E, $45^o$ N). Because we directly compare the L2 retrievals to the L3 product that they create, we filter these based on pixel number and channel-average signal-to-noise ratio (SNR), as is done at the V7 L3 processing stage to improve L3 information content by excluding observations from specific detector elements on MOPITT's detector array that were found to exhibit greater retrieval noise than the other elements (MOPITT Algorithm Development Team, 2017; Deeter et al. 2017).

Specifically, these filters exclude the following: all observations for Pixel 3; and all observations where both (1) the channel 5A SNR < 1000 and (2) the channel 6A SNR < 400. 5A and 6A correspond to the average radiances for MOPITT's length-modulated cell TIR and NIR channels, respectively. Finally, we only retain daytime retrievals, using a solar zenith angle filter of $< 80^o$.

From this subset of L2 retrievals, we take separate area averages for those with a surface index of land

and water, creating two timeseries that are effectively new L3 "land only" and "water only" products, for days when MOPITT retrievals over Halifax are available. We herein refer to these as $L3_L$ and $L3_W$, respectively. For clarity of analysis, we discard remaining L2 retrievals with a surface index of mixed (these account for ~5 % of the total L2 retrieval subset). The number of individual L2 retrievals that are averaged together each day to create $L3_L$ and $L3_W$ is given in Table 1b. From this, it is clear that there are a greater

number of L2 retrievals over water than land within the L3 gridbox containing Halifax, which explains the dominance of water in the $L3_O$ surface classification (Table 1a), and also implies that L2 retrievals over water will have a greater weighting in $L3_O$ than L2 retrievals over land on days when the surface index is mixed.

## 2.2. Retrieval Simulation

To demonstrate how MOPITT retrieved CO concentrations are affected by retrieval sensitivity (Section

3.1.3), we simulate pairs of $L3_L$ and $L3_W$ retrieved profiles that are obtained concurrently (i.e. retrieved on the same day – on some days, one of $L3_L$ or $L3_W$ is missing) as follows:


$$X_{sim} = X_{apr,sim} + A(X_{true,sim} - X_{apr,sim}) \qquad (2)$$

For each simulated retrieval: $X_{true,sim}$ is taken from the Copernicus Atmospheric Monitoring Service (CAMS) reanalysis (CAMSRA – see section 2.3), for the model gridbox that contains Halifax, for the corresponding

month and year of the observed retrieval (because the CAMSRA data are monthly mean values); $X_{apr,sim}$ is the mean of the a priori fields that correspond to the temporally coincident $L3_L$ and $L3_W$ pairings ($APR_{L3-L}$ and $APR_{L3-W}$, respectively); and A is the retrieval averaging kernel from $L3_L$ or $L3_W$. For simulated $L3_L$ ($L3_W$), the averaging kernel from $L3_L$ ($AK_{L3-L}$) ($L3_W$; $AK_{L3-W}$) is used. Thus, any differences between each pair of simulated retrievals ($X_{sim,L3-L}$ and $X_{sim,L3-W}$) are solely a result of differences in A, since $X_{true,sim}$ and

$X_{apr,sim}$ are identical for both. Simulations are initially performed on $\log_{10}(VMR)$ for consistency with the MOPITT retrieval algorithm, and then converted back to VMR scale for analysis.

## 2.3. Additional datasets

The CAMSRA dataset to simulate retrievals is described by Inness et al., (2019). For the CAMSRA gridbox containing Halifax (horizontal resolution = 1º x 1º), we extract CO volume mixing ratios for levels 1000 hPa – 100h Pa at 100h Pa intervals, which correspond to the MOPITT profile levels. The CAMSRA dataset has

no 'surface' profile level, so we take the 1000 hPa profile level (the lowest level available in the dataset) to correspond to MOPITT's floating surface level. At the time of writing, CAMSRA data are only available for the years 2003-2016.

Information on mean wind patterns across Nova Scotia and the surrounding area is taken from the

European Centre For Medium-Range Weather Forecasts (ECMWF) ERA-Interim dataset (horizontal resolution = 0.75º x 0.75º; see Dee et al., (2011) for a dataset overview). We analyse daily mean u and v vector winds for the following levels: 10-metres (the closest level to the surface for winds in the dataset), 850 hPa, and 500 hPa (which correspond roughly to the lower- and mid- troposphere, respectively). In addition, we extract monthly mean temperature profile data (at 100 hPa intervals, plus the skin temperature and 2-

metre air temperature variables) for the closest model gridboxes to Halifax that exclusively cover land and ocean, in order to illustrate the typical "land-only" and "water-only" temperature profiles that correspond to the MOPITT L2 retrievals over land and water that are analyzed.





## 3. Results and Discussion.

### 3.1. Impact of retrieval sensitivity differences on temporally coincident $L3_L$ and $L3_W$ retrievals.

In this section we compare the $L3_L$ and $L3_W$ CO retrievals and demonstrate where and when there are differences in retrieved CO concentrations that are clearly linked to differences in retrieval sensitivity over land and water. We restrict our analysis to days when the $L3_O$ surface index is "mixed" and both $L3_L$ and $L3_W$ retrievals are present, in order to minimize any potential differences in the true profile between land and water (there are a couple of days in the $L3_O$ timeseries when one or the other of $L3_L$ or $L3_W$ is missing, even

when the $L3_O$ surface index is "mixed", owing to the presence of L2 retrievals with a surface index of "mixed", which we have discarded). Thus, an underlying assumption here is that land-water differences in the true profile for retrievals contributing to $L3_L$ and $L3_W$ are small, owing to the fact that they are retrieved in close spatial proximity to eachother (i.e. within the same 1° x 1° gridbox) and at the same time. We test this assumption in Sections 3.1.4 and 3.1.5.

### 3.1.1. Climatology of land-water retrieval and a priori differences.

Figure 2 shows the percentage difference between temporally coincident retrieved VMRs for selected profile levels and for CO total column (TCO) amounts (ΔRET) in $L3_L$ and $L3_W$. Positive (negative) differences indicate that retrieved VMRs/TCO are greater (less) in $L3_W$ than $L3_L$. Differences are expressed as percent values, rather than differences in measurements units, so that we can display profile and TCO retrievals on

the same plot (profile units are ppbv, TCO units are molecules $cm^{-2}$).

In both seasons, mean retrieved VMRs are greater in $L3_W$ than $L3_L$ in the lower troposphere (LT – surface, 900hPa and 800hPa profile levels), with a maximum mean difference in DJF of 11.4 % (14.9 ppbv, p = 0.116) at the 900hPa level, and in JJA of 10.8 % (12.4 ppbv) at the surface level, where the mean difference is highly significant (p = 0.005). The spread of ΔRET values is comparable in both seasons at the surface and 900 hPa

levels (although larger outliers exist in JJA), and much larger at 800 hPa in JJA than DJF, with a clear skew towards positive values in all three cases. This demonstrates that, although on any given day retrieved VMRs in $L3_L$ and $L3_W$ can differ by as much as ~50 % in the LT and especially near to the surface, they are usually greater over water than land in L2 retrievals obtained at the same time within the 1° x 1° L3 gridbox containing Halifax. Mean ΔRET values are closer to zero and generally less significant in the MT and LT (represented

by the 600 hPa and 300 hPa profile levels respectively), indicating that differences in retrieved VMRs are not as persistent at higher altitudes. However, the spread in ΔRET remains large at these altitudes, with retrieved VMRs in $L3_L$ and $L3_W$ differing by over ± 40 % on individual days (with outliers exceeding ± 70





%). TCO is significantly greater in $L3_W$ than $L3_L$ in both seasons (mean difference of 6.2 % in DJF (p < 0.1) and 3.9 % in JJA (p < 0.06)), which is consistent with the most persistent VMR differences occurring in the

LT, where atmospheric densities are greatest, thus contributing a relatively greater amount to the total column than MT and UT profile levels.

Our assumption in this section is that L2 retrieved CO concentrations obtained within the same 1º x 1º L3 gridbox should be similar. We may actually expect retrieved CO amounts in $L3_L$ to be greater than those in $L3_W$ due to CO sources existing on land, particularly within the city of Halifax. One reason for the ΔRET

values instead indicating higher concentrations over water could be differences in the a priori profiles (ΔAPR) used in the corresponding retrievals. The L2 retrievals over land and water have different a priori profiles owing to spatial interpolation of the 1º x 1º model climatology to the 22 x 22 km footprint of the MOPITT L2 retrieval. However, as Figure 2 demonstrates, ΔAPR values are small in comparison to ΔRET, with mean difference values very close to zero and a maximum range of 9.5 % (occurring at the surface level in JJA).

Moreover, the sign of mean ΔAPR does not match that of mean ΔRET at several levels in both seasons (i.e. retrieved VMRs in $L3_W$ are greater than in $L3_L$, but a priori VMRs are less). It therefore appears unlikely that a priori profile differences are responsible for the observed differences in retrieved CO concentrations in $L3_L$ and $L3_W$.

### 3.1.2. Climatology of land-water retrieval sensitivity differences.

An alternative explanation for the observed ΔRET could be differences in retrieval sensitivity over land and water, quantified by the retrieval AK matrix. Figure 3 compares the mean AKs corresponding to the retrieved profiles in $L3_L$ and $L3_W$ analysed in the previous section. Each curve corresponds to a row of the AK matrix, with the width of each AK giving a measure of the vertical resolution associated with a specific level of the retrieved profile; the widest part of each AK (when a peak is evident) indicates the portion of the true

profile that the corresponding level of the retrieved profile is most sensitive to. The sum of the elements in each AK row ($AK_{rowsum}$) represents the overall sensitivity of the retrieved profile at the corresponding pressure level to whole the true profile; values close to zero indicate that the retrieval is relatively insensitive to the true profile and therefore closely tied to the a priori profile, while the converse is true as the rowsum approaches one. The mathematical trace of the AK matrix (i.e. sum of the diagonals) gives the degrees of

freedom for signal (DFS) of the retrieval, which is a measure of the number of independent pieces of information (in other words, "information content") in the retrieval from the measurement, with respect to the true profile. When DFS values approach two, this is interpreted as the retrieval being able to resolve CO in two independent atmospheric layers.




There are some clear differences between the mean AKs over land (from $L3_L$) and water (from $L3_W$)
shown in Figure 3. In DJF, AKs for LT (especially surface level) retrievals reach greater values in the LT in
$L3_W$ than $L3_L$, indicating that sensitivity to the true profile at these levels is actually greater in $L3_W$ (surface
level AK peak = ~0.15 for water, ~0.10 for land, both at the surface). This is reflected in greater rowsum
values for LT AKs in $L3_W$ than $L3_L$ (0.55 vs 0.44 for surface level AKs, to 0.82 vs 0.74 for 800 hPa level
AKs, respectively). Differences in MT (600 hPa) and UT (300 hPa) AKs are much less pronounced, with
sensitivity to the true profile actually becoming slightly greater in $L3_L$ than $L3_W$ higher up in the troposphere.
In JJA, the mean LT and MT AKs are qualitatively much more different between $L3_L$ and $L3_W$ than in DJF.
LT rowsums are significantly greater in $L3_L$ ($p < 0.001$ at the surface, 900 hPa and 800 hPa profile levels,
see Supplementary Material 1) and closer to one, signifying that these retrievals contain much more true
profile information than those in $L3_W$ ($AK_{rowsums}$ = 0.15 vs 0.70 at the surface level to 0.67 vs 0.94 at 800
hPa, in $L3_W$ and $L3_L$ respectively). The AK shapes also indicate that the respective retrievals are sensitive to
different parts of the true profile. In $L3_L$, surface and LT AKs peak at either the surface (surface level AK)
or at 900 hPa (both the 900 hPa and 800 hPa level AKs) and decline towards the UT, while MT AKs indicate
relatively equal sensitivity throughout the LT, MT and the lower levels of the UT. In $L3_W$ on the other hand,
LT and MT AKs (excluding the largely insensitive surface level) indicate relatively little sensitivity in the
lowest profile levels and peak in the MT. The surface level AK in $L3_W$ actually indicates close to zero
sensitivity throughout the profile ($AK_{rowsum}$ = 0.15), aside from a very weak peak at the surface level, which
is around 3 times lower than that over land. As in DJF, UT AKs are quite similar, except for relatively small
differences in the LT. Differences in mean DFS values for retrievals in $L3_L$ and $L3_W$ are greater in JJA (-
0.11) than DJF (0.07), highlighting the greater land-water sensitivity contrast in JJA but also reflecting the
switch in surface type exhibiting the greatest LT retrieval sensitivity between the seasons.

The differences in surface and LT AKs for MOPITT retrievals in $L3_L$ and $L3_W$ discussed above can be
accounted for by the differing LT thermal contrast conditions over land and water, as explored in detail by
Deeter et al. (2007). The sensitivity of MOPITT retrievals to CO in the LT is predominantly controlled by
the thermal contrast between the surface skin temperature ($T_{skin}$) and the surface air temperature ($T_{sfc}$), and
by the tropospheric temperature profile. Seasonal mean temperature profile data from ERA-Interim for the
nearest land-only and water-only model gridboxes to Halifax show clear differences in DJF and JJA (Figure
4). In DJF, $T_{skin}$ is around 6° K warmer than the 2-meter air temperature ($T_{2m}$, which we use as a proxy for
$T_{sfc}$ as this is the lowest model level) over water, and a further degree warmer than the air at 1000 hPa,
whereas the temperature gradient is weak/slightly inverted over land, with $T_{skin}$ less than a degree warmer
than $T_{2m}$, which is actually slightly cooler than the air at 1000 hPa. Correspondingly, the lowest couple of
retrieval levels indicate greater sensitivity to the surface and LT over water (in $L3_W$) than land (in $L3_L$) in





DJF (Figure 3). Temperature profiles converge towards the MT, as do AKs. In JJA on the other hand, there is a clear gradient between $T_{skin}$ and the overlying air on land, while the ocean surface is actually cooler than the air above, up to a height of 900 hPa. As a result, surface and LT sensitivity is greater in $L3_L$ than in $L3_W$

in JJA, with the sensitivity of retrievals in $L3_W$ approaching zero close to the surface owing to this inverted temperature profile. The $T_{skin}$ increase approaching $20^o$ K  likely also accounts for the relatively greater overall true profile sensitivity in $L3_L$ in JJA than in $L3_W$ in DJF.

That retrieval sensitivity contrast between $L3_L$ and $L3_W$ is most pronounced in the LT is consistent with the finding that retrieved CO profiles in $L3_L$ and $L3_W$ show the greatest contrast in the LT (Section 3.1.1,

Figure 2). In DJF, the retrieval is more sensitive to the true profile in the LT in $L3_W$ (Figure 3, top row) and retrieved LT VMRs are greater in $L3_W$ than $L3_L$. In JJA, the retrieval is more sensitive to the true profile in the LT in $L3_L$ (Figure 3, bottom row), and retrieved LT VMRs are less in $L3_L$ than $L3_W$. This implies that in DJF, LT CO retrievals in $L3_L$ are weighted more heavily towards an a priori profile in which LT VMRs are lower than in the true CO profile; while in JJA, LT retrievals in $L3_W$ are weighted more heavily towards an

a priori profile in which VMRs are greater than in the true profile. We explore this point in detail in Section 3.2.2. Although mean retrieval sensitivity in $L3_L$ and $L3_W$ converges with altitude, differences do exist from day-to-day but they are neither as large, nor as skewed in favour of retrievals over land or water (depending on season) as in the LT, where there is a well understood thermal contrast mechanism creating the systematic land-water sensitivity contrast (see Supplementary Material 1). Likely causes for this could be day-to-day

changes in atmospheric conditions (i.e. temperature or water vapour profiles), or random instrumental or retrieval noise.

### 3.1.3. Control of ΔRET by land-water sensitivity differences

To demonstrate that retrieval sensitivity differences over land and water can lead to the observed differences between CO profiles in $L3_L$ and $L3_W$ that are retrieved at the same time, we simulate and compare the pairs

of retrieved CO profiles in $L3_L$ and $L3_W$ that are analysed in the preceding sections, using the transformation outlined in Section 2.2. For each $L3_L$- $L3_W$ simulated retrieval pairing ($X_{sim,L3-L}$ and $X_{sim,L3-W}$), the true and a priori profiles are common to both (recall the negligible mean difference between $L3_L$ and $L3_W$ a priori profiles in Figure 2), but the retrieval AKs from $L3_L$ and $L3_W$ are used (i.e. for simulated retrievals in $L3_L$ ($L3_W$), the AK from $L3_L$ ($L3_W$) is used). Thus, any differences between each pair of simulated retrievals are

solely a result of differences in AKs. Owing to available CAMSRA data only covering the years 2003-2016, as opposed to MOPITT data being available for 2000-2017, only a (large) subset of the retrieval pairings considered in earlier sections could be simulated.


We first demonstrate the sensitivity effect with a case study on 2013-08-18. For brevity, we focus only on the simulated surface level retrievals ($X_{sim,L3-L,sfc}$ and $X_{sim,L3-W,sfc}$). Profiles (a priori ($X_{apr,sim}$) and truth

($X_{true,sim}$)) and surface level AKs ($AK_{sfc}$) used in the simulation, and the resulting $X_{sim,L3-L}$ and $X_{sim,L3-W}$, are given in Table 2. $AK_{sfc}$ from $L3_L$ and $L3_W$ are also shown in Figure 5. In this example, $X_{true,sim}$ is considerably lower than $X_{apr,sim}$ at the surface level (-60.36 ppbv; $X_{true,sim,sfc}$ and $X_{apr,sim,sfc}$ respectively) and several features of the AKs indicate greater sensitivity to $X_{true,sim}$ in $L3_L$ and $L3_W$: the surface component of $AK_{sfc}$ ($AK_{diag,sfc}$) is significantly greater in $L3_L$ than $L3_W$ (0.23 vs 0.01), and the rowsum ($AK_{rowsum,sfc}$) is over 5 times as high

(0.84 vs 0.15). Correspondingly, $X_{sim,L3-L,sfc}$ is 24.92 ppbv lower than $X_{sim,L3-W,sfc}$ and much closer to $X_{true,sim,sfc}$ (32.21 ppbv higher vs 57.13 ppbv higher). Both $X_{sim,sfc}$ values indicate that $X_{true,sim,sfc}$ is lower than $X_{apr,sim,sfc}$, but $X_{sim,L3-L,sfc}$ gives the closer estimate, as would be expected from the greater $AK_{diag,sfc}$ and $AK_{rowsum,sfc}$ values from $L3_L$. In both cases, a portion of the overall $X_{sim,sfc}$ departures from $X_{apr,sim,sfc}$ (in other words, "value added" over the a priori) originates at other profile levels, and not the surface. This is a result

of $AK_{sfc}$ being nonzero at other profile levels, and is a function of the inter-level correlation of the original retrieval, which is linked to the a priori covariance matrix used in the retrieval (Deeter et al., 2010). Because $AK_{sfc}$ has a well-defined peak at the surface $L3_L$ (Figure 5), while $AK_{sfc}$ over $L3_W$ actually indicates greatest retrieval sensitivity (albeit very weak) to a broad region from $900 - 400$ hPa, the surface accounts for a greater proportion of the $X_{sim,L3-L,sfc}$ departure from $X_{apr,sim,sfc}$ (55 % vs 25 % – see Appendix B for full derivation of

these values) while the $X_{sim,L3-W,sfc}$ departure from $X_{apr,sim,sfc}$ has a greater contribution from the lower-to-mid troposphere as a whole.

We now consider differences between all $X_{sim,L3-W}$ and $X_{sim,L3-L}$ pairings (ΔSIM) throughout the profile and compare these to the observed differences between temporally coincident retrievals in $L3_L$ and $L3_W$ (ΔRET) discussed previously (Figure 6). For ease of comparison, ΔRET values are overlaid (faint lines) for

the shorter time period matching CAMSRA data availability (the 2003-2016 ΔRET patterns are very similar to those seen in Figure 2). It should be noted that we cannot expect ΔSIM to match ΔRET exactly, owing to (possibly large) differences between the $X_{true,sim}$ profiles used in the simulations and the (unknown) true profiles at the time of the actual MOPITT retrievals ($X_{true}$). For instance, $X_{true,sim}$ is a monthly mean value from a reanalysis model, whereas $X_{true}$ varies by day, which should result in less variance in ΔSIM than

ΔRET. In DJF, mean ΔSIM is negligible at all profile levels shown, and the range of values is far smaller than seen for ΔRET. In JJA on the other hand, ΔSIM reaches considerably larger values than in DJF and mean values are significantly different ($p < 0.05$) over land and water at all but one profile level shown (300 hPa). LT and MT ΔSIM distributions are a remarkably good match for ΔRET, given the likely substantial differences between $X_{true,sim}$ and $X_{true}$. In the UT, ΔSIM values are somewhat smaller, although this isn't

unexpected given the smaller land-water AK differences evident in Figure 3.





That $L3_L$ and $L3_W$ simulated and retrieved profile differences over land and water are comparable in JJA is clear evidence that $\Delta RET$ is strongly influenced by the land-water sensitivity contrast in summer months. However, while $\Delta SIM$ values are of a much smaller magnitude in DJF than in JJA, $\Delta RET$ is actually of a similar magnitude in both seasons. The question therefore arises as to why $\Delta SIM$ is so different from $\Delta RET$ in DJF, and why it is of much smaller magnitude in DJF and JJA. Considering the terms of Equation 2, there are two possible explanations. Firstly, differences between $L3_L$ and $L3_W$ LT and MT AKs (A in Equation 2) are much smaller in DJF than in JJA (Figure 3). This means that the deviation of $X_{sim,L3-W}$ and $X_{sim,L3-L}$ from $X_{apr,sim}$ will be more similar in DJF (resulting in small $\Delta SIM$), as opposed to in JJA where, as was seen in the case study discussed above, $X_{sim,L3-L}$ can deviate more from $X_{apr,sim}$ than $X_{sim,L3-W}$ owing to increased sensitivity in $L3_L$ in the lower profile levels (resulting in greater $\Delta SIM$ than in DJF). In the UT, AK differences are comparable in both seasons, and $\Delta SIM$ is correspondingly similar. Secondly, the magnitude of $X_{true,sim} - X_{apr,sim}$ from Equation 2 is, on average, around 4 (3) times greater in JJA than in DJF at the surface (900 hPa) level (Figure 6b). $X_{sim,L3-L}$ therefore deviates more from $X_{apr,sim}$ than $X_{sim,L3-W}$ throughout the LT and MT in JJA owing to strong contrasts in near-surface sensitivity at these profile levels, thus yielding large $\Delta SIM$. Conversely, closer $X_{true,sim}$ and $X_{apr,sim}$ profiles combine with small sensitivity differences to limit $\Delta SIM$ in DJF. A final, alternative explanation for $\Delta SIM$ being a poor match for $\Delta RET$ in DJF, unlike in JJA, is that sensitivity differences in $L3_L$ and $L3_W$ could have less of an impact on $\Delta RET$ in DJF than in JJA, and that something else is responsible – for example, "real" differences between the true CO profile over land and water. This is something we explore in detail in the following sections.

### 3.1.4. Regional land-water contrast in L2 data

To further evaluate whether $\Delta RET$ within the L3 gridbox containing Halifax is a function of land-water sensitivity contrasts, or actually due to real gradients in the true CO profile (for example, due to offshore transport of emissions from the city of Halifax or marine-land chemistry differences), we analyse the characteristics of retrieved profiles over the broader geographical region containing Halifax. If $\Delta RET$ is linked to sensitivity, then we would expect there to be a clear land-sea contrast across the whole region. This is exactly what we see. Figure 7 shows seasonal median L2 retrieved and a priori CO concentrations for the surface profile level, where $\Delta RET$ and $L3_L$-$L3_W$ AK differences are greatest and most significant, for the Canadian maritime provinces and a small portion of northeast USA (note that we show seasonal median fields here as the spatial patterns are clearer than for plots consisting only of the subset of days analysed in the rest of this section. The corresponding plot for the subset of days is shown in Supplementary Material 2, and the main findings are unchanged). Difference fields (RET-APR) are also shown. In JJA, a land-sea



contrast is remarkably clear in both the retrieved and RET-APR fields. The a priori field shows elevated CO amounts emanating from the west-southwest (indicative of CO sources in northeast USA and around the Great Lakes) and decreasing quite smoothly towards the north and east. The west-southwest maxima is

replicated in the retrieved field, but the smoothly decreasing gradient is clearly broken, with the land in the image characterized by lower CO values than the adjacent ocean. This contrast is enhanced further in the RET-APR field. RET-APR values close to zero indicate either very low retrieval sensitivity or closely matching retrieved and a priori values. The analysis of averaging kernels in Section 3.1.2 demonstrated the lack of retrieval sensitivity at the surface over water (in $L3_W$) in JJA. On average, RET-APR is over 10 ppbv

lower over land than over water (determined by binning the L2 data for the region shown according to surface classification), a highly significant difference (p = 0.000, from 2-tailed T-test of binned data). This reinforces our earlier interpretation that, in JJA, LT retrievals in $L3_W$ are weighted more heavily towards an a priori profile in which CO concentrations are too high than LT retrievals in $L3_L$.

The land-sea contrast is less clear in DJF, although the RET-APR field does indicate generally positive

values over water and negative values over land. The contrast being less apparent in DJF compared to JJA is consistent with the smaller $L3_L$-$L3_W$ AK differences in DJF compared to JJA. However, it is surprising that RET-APR changes sign from land (generally negative) to water (generally positive). This may be linked to some factor other than retrieval sensitivity to the true CO profile, such as errors in retrieved/a priori surface temperatures or emissivities, which are important components of the radiative transfer model used in

MOPITT's CO retrieval algorithm. If this were the case over the sea, it could explain the maxima in RET-APR values to the northwest of Halifax in the Bay of Fundy, where the water is relatively shallow and the tidal range is the highest in the world at 16.3 m, transporting large amounts of suspended sediments (which will affect emissivity). Alternatively, the difference could reflect a physical process causing elevated CO over the ocean, although this seems unlikely, as we expect atmospheric transport to minimize such a contrast.

Corresponding maps for selected other retrieval levels and TCO are shown in Supplementary Material 3. In JJA, a land-sea contrast is qualitatively evident at all profile levels and for TCO, with the exception of 800 hPa; and in DJF a contrast is evident at 800 hPa and for TCO.

### 3.1.5. Can ΔRET be explained by circulation-driven horizontal gradients in the TRUE CO profile?

The preceding sections presented evidence that differences in temporally coincident $L3_L$ and $L3_W$ retrieved

profiles are linked to sensitivity contrasts over land and water, especially in JJA and in the LT. However, these differences could also be a result of horizontal gradients in the true CO profile. It is plausible that retrieved LT VMRs are greater in $L3_W$ than in $L3_L$ due to, e.g., offshore transportation of CO by regional winds either from Halifax or from the large polluting areas on the northeast coast of the US and around the





Great Lakes (as seen in the general decline in CO amounts from the west-southwest towards the northeast in
Figure 7). Winds generally tend from the west/northwest/southwest in this area (Figure 8), which will lead
to offshore transport of continental pollution.

We compare composite mean wind patterns across Nova Scotia using ERA-Interim data for days when
retrieved surface level VMRs in $L3_W$ are greater than in $L3_L$ ($L3_W > L3_L$) and days when they are less ($L3_W$
$< L3_L$) to illustrate the likely atmospheric transport direction of CO (and other atmospheric constituents) in
the region. These are shown in Figure 8 (10-metre winds are used). There is a clear circulation difference
evident in DJF. When $L3_W > L3_L$ (13 days of the 19 in Figure 2) the wind is in an offshore direction, whereas
when $L3_W < L3_L$ (6 days of the 19 in Figure 2) the wind is alongshore from the west-southwest (and
noticeably weaker). This is evidence to suggest that the LT ΔRET patterns in DJF are linked to the horizontal
gradients in the true CO profile, although the small sample sizes involved here dictate caution. In contrast to
DJF, however, there is no clear circulation difference in JJA; the winds are generally from the west-southwest
irrespective of whether $L3_W > L3_L$ or $L3_W < L3_L$. This lends further support to the conclusion that LT ΔRET
in JJA is strongly linked to the demonstrated land-water sensitivity contrast. The seasonal difference in these
results could explain why LT (and especially near-surface) ΔRET is of comparable magnitudes in DJF and
JJA (Section 3.1.1), despite smaller $L3_L$-$L3_W$ retrieval sensitivity contrasts and ΔSIM values in DJF than JJA
(Sections 3.1.2 and 3.1.3 respectively). In other words, in DJF, ΔRET is more indicative of differences in
true CO concentrations, while in JJA it is more strongly tied to differences in retrieval sensitivity. This is not
to say contrasts in retrieval sensitivity do not influence LT ΔRET in DJF; just that the effect is not as strong
as in JJA, when the LT sensitivity contrast is greater. We only consider the surface profile level here; the
findings are consistent higher up in the LT, but the DJF circulation difference is absent in the MT (see
Supplementary Material 4), consistent with ΔRET being much smaller in the MT and above, on average.

While there is no obvious circulation difference at the surface in JJA between days when $L3_W > L3_L$ and
$L3_W < L3_L$, there is evidence that these days actually cover different parts of the analysed MOPITT
timeseries, which spans 17 JJA seasons from 2000 to 2016. Days when $L3_W < L3_L$ tend to fall towards the
start of the timeseries (mean year in which the retrieval was performed = 2004), whereas days when $L3_W >$
$L3_L$ tend to be later (mean retrieval year = 2008). The same is not true in DJF (mean retrieval years = 2008
and 2007 respectively). As we show in Section 3.2.2, this appears to be a function of true LT CO
concentrations over Halifax decreasing across the time period studied, which is detectable in JJA by the
retrievals in $L3_L$ owing to their greater near-surface sensitivity, but not by the retrievals in $L3_W$, which are
tied to an a priori value that is too low at the start of the timeseries and too high at the end.



**3.2. Consequences for $L3_O$ timeseries**

In this section, we demonstrate the impact that sensitivity driven differences in L2 retrieved profiles have on the statistics of the original L3 timeseries for Halifax ($L3_O$) that they are used to create, and consequences therein for the results of a typical temporal trend analysis. We do this through comparison with the $L3_L$ and $L3_W$ timeseries. For clarity, unlike in the previous section, the analysis in this section is not only limited to 505 days when the $L3_O$ surface index is mixed; all days are considered.

**3.2.1. Impact on seasonal data distribution**

In both DJF and JJA, retrieved CO VMRs at the surface in $L3_O$ share a distribution that is qualitatively much more similar to $L3_W$ than $L3_L$, with smaller (and less statistically significant) mean differences (Figure 9). This is to be expected: the $L3_O$ timeseries consists of a greater proportion of L2 retrievals over water than 510 over land (Table 1), primarily because more of the surface within the L3 gridbox containing Halifax is water than land (owing to the location of the coastline – see Figure 1). Thus, there is a greater likelihood of the $L3_O$ surface index being classified as water (and only L2 retrievals over water being retained for the area averaging in L3 data creation), and also of more L2 retrievals over water contributing to the area average than L2 retrievals over land when the $L3_O$ surface index is mixed (see Section 2.1.2).

The difference in the distribution of surface level retrieved CO VMRs between $L3_L$ and both of $L3_O$ and $L3_W$ is most striking in JJA. This is to be expected, given that this is when the $L3_L$-$L3_W$ sensitivity differences are greatest. Although the mean difference between $L3_O$ and $L3_L$ is relatively small (5.9 ppbv, p = 0.04), the spread of values is far greater for $L3_L$ (standard deviation (interquartile range) = 35.4 ppbv (44.3 ppbv)) than for $L3_O$ (standard deviation (interquartile range) = 19.1 ppbv (19.7 ppbv)). This lower variability in $L3_O$ is a 520 direct result of the fact that L2 surface level retrievals over water, which have a significantly lower information content than L2 surface level retrievals over land in the summer months and are therefore more closely tied to the a priori CO concentration (Section 3.1.2), have a greater contribution to the $L3_O$ timeseries than L2 retrievals over land. $L3_L$ contains significantly more information on the true CO concentration variations at the surface (mean surface level $AK_{rowsum}$ = 0.70 and 0.25 for $L3_L$ and $L3_O$, respectively, p = 525 0.000) – albeit at the cost of over 55 % fewer observations being available for analysis during the time period covered (n = 106 for $L3_L$ vs 248 for $L3_O$) – yet this information on CO variability is lost to users of the $L3_O$ timeseries because of the way the L3 products are created, as described in Section 2.1.2. The same is not true in DJF. Although $L3_W$ has marginally more information on the true CO concentrations at the surface than $L3_O$ (mean surface level $AK_{rowsum}$ = 0.55 and 0.54, respectively), which has its information content degraded 530 slightly by the influence of L2 retrievals over land (mean surface level $AK_{rowsum}$ = 0.44), the difference in





CO concentrations in the two timeseries is negligible and not statistically significant (1.8 ppbv, p = 0.49), owing to the dominance of L2 retrievals over water on the $L3_O$ timeseries. Thus, unlike in JJA, there is no significant loss of information on the true CO VMR variations at the surface for users of the $L3_O$ timeseries. It is also noticeable that the spread of values is much more similar across the three timeseries than in JJA

(standard deviation (interquartile range) = 24.9, 22.2, 19.9 (25.2, 25.0, 24.2) for $L3_L$, $L3_W$ and $L3_O$, respectively), suggesting a lack of systematic difference in the values that are being retrieved, in contrast to JJA.

Although the sample sizes considered here are different, because L2 retrievals over land and water are not necessarily always present on the same days (e.g. due to variable cloud coverage), these results hold if

the preceding analysis is restricted to days when $L3_L$ and $L3_W$ are both present (see Supplementary Material 5). We only present analysis for the surface profile level here as this is where $L3_L$-$L3_W$ differences in retrieved VMRs and retrieval sensitivity are greatest. Plots of other profile levels and TCO are given in Supplementary Material 6.

### 3.2.2. Consequences for temporal trend analysis

Results from Ordinary Least Squares (OLS) regression analyses on seasonal mean profile and TCO values from the three different timeseries are presented in Table 3. OLS best-fit lines, along with boxplots of seasonal data distributions, are shown in Figure 10 for the surface profile level, where the $L3_L$-$L3_W$ difference in retrieval sensitivity and retrieved VMRs is greatest. In our analysis we focus on the gradient parameter of the best-fit line ($m_{L3-O}$, $m_{L3-W}$, and $m_{L3-L}$), since this describes the trend in seasonal mean values over time.

In general, the trend direction is consistent across all 3 timeseries in both DJF and JJA at all profile levels considered (with the exception of 600 hPa in JJA, discussed below) and for TCO. The trend magnitudes $m_{L3-O}$ and $m_{L3-W}$ are quite closely matched, while $m_{L3-L}$ is most often different from both. This is not surprising, given the greater influence of L2 retrievals over water on the $L3_O$ timeseries, as explained earlier.

At the surface in JJA (Figure 10), $m_{L3-L}$ shows a decrease ~6 times greater than $m_{L3-W}$ and almost 3 times

greater than $m_{L3-O}$. The difference between $m_{L3-L}$ and both $m_{L3-W}$ and $m_{L3-O}$ is highly statistically significant (p < 0.01) in both cases. This is because $L3_W$ lacks the retrieval sensitivity in JJA to detect a change in the true CO profile close to the surface and is tied to the a priori which has no yearly change. Consequently, $m_{L3-W}$ and, by extension, $m_{L3-O}$ significantly underestimate the declining trend in surface level CO in JJA that $m_{L3-L}$ is able to detect owing to the greater retrieval sensitivity over land during this season. The trend

underestimation persists at 900 hPa, albeit more weakly ($m_{L3-L}$ is now only ~2 times greater than $m_{L3-W}$ and $m_{L3-O}$), but the differences remain statistically significant (p < 0.1). By 800 hPa, this trend underestimation in $m_{L3-W}$ and $m_{L3-O}$ has all but gone away, likely owing to the fact that the retrieval in $L3_W$ contains a moderate


amount of true profile information at this level. In DJF, trends at LT profile levels are far more similar across the three timeseries compared to JJA, owing to the much smaller retrieval sensitivity differences between

$L3_L$ and $L3_W$; the differences are largest at the surface profile level (but not statistically significant), and by 800 hPa the trends have almost converged, consistent with the DJF AK comparison (Figure 3, Section 3.1.2).

At 600 hPa in JJA, $m_{L3-O}$ and $m_{L3-W}$ indicate a decrease of 0.55 and 0.86 ppbv $y^{-1}$ respectively, while $m_{L3-L}$ is effectively zero, with no statistical evidence that any trend exists (p = 0.942). While at this profile level the overall information content of the retrievals is effectively the same in $L3_L$ and $L3_W$ (indicated by AK

rowsums 1.03 and 1.01 respectively (Figure 3)), the AK shape is completely different: retrieval sensitivity in $L3_W$ has a strong peak at 600 hPa, as opposed to the retrieval sensitivity in $L3_L$ being uniform throughout the LT, MT and into the UT. Therefore, $m_{L3-W}$ and, by extension, $m_{L3-O}$, appear to be more indicative of the actual temporal trend in the MT than $m_{L3-L}$, which seems to be detecting a mixed signal between decreasing CO in the LT-MT and increasing CO in the UT (see below), owing to the broad sensitivity profile of the $L3_L$

retrieval at this level. The influence of L2 retrievals over land on the $L3_O$ timeseries thus clearly weakens $m_{L3-O}$ in JJA, which loses statistical significance and indicates a decrease in CO that is ~36 % weaker than the more reliable $m_{L3-W}$. As in the LT, in DJF the trends at 600 hPa are highly similar, which is to be expected given the closely-matched AKs for retrievals in $L3_L$ and $L3_W$ at this profile level.

Trends at 300 hPa also display differences between the $L3_O$, $L3_L$ and $L3_W$ timeseries. In DJF, $m_{L3-L}$

indicates a slower increase in VMRs than $m_{L3-W}$ and $m_{L3-O}$, and confidence that this trend is statistically different from zero is low, unlike for the trends in the other two timeseries. In JJA, $m_{L3-L}$ indicates a faster increase in VMRs than $m_{L3-W}$ and $m_{L3-O}$, but the differences in trend magnitude are not statistically significant (p > 0.1). Causes for these trend differences are difficult to establish given that 300 hPa AKs are similar in $L3_L$ and $L3_W$ in both seasons. It is plausible, however, that subtle AK differences, especially at LT profile

levels in JJA, will influence the retrieved values enough to impact upon the trends detected in this analysis, especially if true CO concentrations are changing by > ±1 % $yr^{-1}$ throughout the profile as this analysis suggests.

In comparison to trends at most profile levels, differences in TCO trends between the three timeseries are relatively small in both seasons. This could be because TCO retrievals are better constrained than retrievals

for individual profile levels, whose retrieved VMRs depend upon the true profile sensitivity of the retrieval. The weakness and lack of statistical significance of TCO trends in JJA is possibly a result of the strong UT increase in CO offsetting the LT and MT decrease. On the other hand, the stronger and statistically significant (p < 0.1) decrease in TCO in DJF may be due to UT increases being weak in comparison to LT decreases. It should be borne in mind that, as neither season has DFS > 2 (Figure 3), we cannot be entirely confident about

the vertical location of the trends identified in this analysis since the retrievals lack the vertical resolution to



detect multiple atmospheric layers independently (although confidence is greater in JJA than in DJF, owing to greater DFS values over both surfaces in summer).

To test whether the reported trend differences occur as a result of $L3_O$, $L3_L$ and $L3_W$ sometimes sampling different days, we restricted the OLS regression analyses to days when all three datasets are present. The

main results hold, although the regression parameters are more poorly constrained (i.e. larger standard error and confidence intervals, less statistical significance) owing to the smaller sample sizes. Results from this restricted analysis are discussed further in Supplementary Material 7.

## 4. Conclusions.

Users of MOPITT products are advised to filter the data before analysis in order to maximize the influence

of satellite measurements and minimize the impact of a priori CO concentrations on results. In particular, it is advised that retrievals over water, which are known to have lower information content than retrievals over land, are discarded. This is especially so for the analysis of temporal trends in CO concentrations, owing to the year-to-year stationarity of the a priori. However, for L3 gridboxes that straddle the coastline, the ability to apply such filtering is limited since the products will generally have some contribution from L2 retrievals

that take place over both land and water. This is a direct consequence of the way that L3 products are created.

As we have explicitly demonstrated for the 1° x 1° L3 gridbox containing the coastal city of Halifax, Canada, the L2 retrieved CO concentrations, from which the L3 products are created, differ depending on whether the retrieval took place over land or water. In JJA, and especially near to the surface, this is directly linked to differences in the sensitivity of the retrievals to the true CO profile. The merging of these retrievals

to create the L3 product can significantly affect the statistics of the dataset and the results of temporal trend analysis with the data, with the largest and most statistically significant effects at the surface, where land-water sensitivity contrasts are greatest. As we show, results that are more representative of changes of true CO concentrations close to the surface within the L3 gridbox containing Halifax can only currently be obtained by use of the L2 products, which can be filtered by surface type to maximize information content.

Our results suggest that L2 retrievals over land and water should not both contribute to L3 products in coastal gridboxes. This is consistent with previous data filtering recommendations (MOPITT Algorithm Development Team, 2017; Deeter et al., 2015). The horizontally averaged $L3_L$ and $L3_W$ timeseries that we have analysed in this paper are effectively L3 "land-only" and L3 "water-only" datasets, and these offer an alternative in this respect that preserves the benefits of available L3 products – namely, less computing

resources and expertise required for their analysis compared to L2 products, which broadens access to the data – but offers users the flexibility to select over which surface the contributing retrievals were performed in order to maximize the information content of L3 data in coastal gridboxes. Although our study has only



focused on the city of Halifax, the results are of relevance to all coastal L3 gridboxes, which contain 6 of the top 10 and 43 of the top 100 agglomerations by population and are therefore likely targets for analysis of

temporal changes in air pollution indicators such as CO, especially near to the surface. The degree of information content limitation in the L3 data will depend on the relative contributions of L2 retrievals over land and water to each specific L3 gridbox, as well as on the strength of the land-water retrieval sensitivity difference, which in turn depends on scene-specific geophysical variables such as surface temperature and emissivity. Work is currently ongoing to compare the results of analyses conducted with MOPITT L2 and

L3 CO data over these cities.

**Data Availability**

MOPITT data were downloaded from the NASA Earthdata portal (https://search.earthdata.nasa.gov/). CAMSRA and ERA-Interim data were downloaded from the ECMWF public datasets portal (https://apps.ecmwf.int/datasets/).

**Appendix A: List of acronyms**

| | |
|---|---|
| $\Delta RET$ | Percentage differences in retrieved CO concentrations in $L3_L$ and $L3_W$ |
| $\Delta SIM$ | Percentage differences in simulated retrieved CO concentrations in $L3_L$ and $L3_W$ |
| AK | Averaging kernel |
| $AK_{L3-L}$ | Averaging kernel from $L3_L$ |
| $AK_{L3-W}$ | Averaging kernel from $L3_W$ |
| $APR_{L3-L}$ | A priori from $L3_L$ |
| $APR_{L3-W}$ | A priori from $L3_W$ |
| CAM-chem | Community Atmosphere Model with Chemistry |
| CAMS | Copernicus Atmospheric Monitoring Service |
| CAMSRA | CAMS Reanalysis |
| CO | Carbon Monoxide |
| DFS | Degrees of freedom for signal |
| DJF | December, January, February |
| JJA | June, July, August |
| L2 | Level 2 data products |
| L3 | Level 3 data products |





| | | |
|---|---|---|
| L3$_L$ | | Area-averaged L2 data for the L3 gridbox containing Halifax, for L2 retrievals over land only |
| L3$_O$ | | Original, 'as-downloaded' L3 timeseries for the 1o x 1o L3 gridbox containing Halifax |
| L3$_W$ | | Area-averaged L2 data for the L3 gridbox containing Halifax, for L2 retrievals over water only |
| LT | | Lower troposphere (surface – 800hPa profile levels in MOPITT data) |
| MERRA-2 | | NASA Modern-Era Retrospective Analysis for Research and Applications V2 |
| m$_{L3-L}$ | | Gradient parameter of best-fit line calculated from OLS regression analysis on seasonal mean CO concentrations from L3$_L$ timeseries |
| m$_{L3-O}$ | | Gradient parameter of best-fit line calculated from OLS regression analysis on seasonal mean CO concentrations from L3$_O$ timeseries |
| m$_{L3-W}$ | | Gradient parameter of best-fit line calculated from OLS regression analysis on seasonal mean CO concentrations from L3$_W$ timeseries |
| MOPITT | | Measurement of Pollution in the Troposphere (instrument) |
| MT | | Mid-troposphere (700hPa – 500hPa profile levels in MOPITT data) |
| NIR | | Near infrared |
| SNR | | Signal-to-noise ratio |
| TCO | | Total Column CO |
| TIR | | Thermal infrared |
| T$_{sfc}$ | | Surface air temperature |
| T$_{skin}$ | | Surface skin temperature |
| UT | | Lower troposphere (400hPa – 100hPa profile levels in MOPITT data) |
| V7 | | MOPITT Version 7 data products |
| VMR | | Volume Mixing Ratio |
| X$_{apr}$ | | A priori CO VMR profile |
| X$_{apr,sim}$ | | A priori CO VMR profile used in retrieval simulation |
| X$_{apr,sim,sfc}$ | | A priori surface profile level CO VMR  used in retrieval simulation |
| X$_{rtv}$ | | Retrieved CO VMR profile |
| X$_{sim}$ | | Simulated retrieved CO VMR profile |
| X$_{sim,L3-L}$ | | Simulated retrieved CO VMR profile from L3$_L$ |
| X$_{sim,L3-L,sfc}$ | | Simulated retrieved surface profile level CO VMR from L3$_L$ |
| X$_{sim,L3-W}$ | | Simulated retrieved CO VMR profile from L3$_W$ |





$X_{sim,L3-W,sfc}$       Simulated retrieved surface profile level CO VMR from $L3_W$

     $X_{true}$            'True' CO VMR profile

     $X_{true,sim}$        'True' CO VMR profile used in retrieval simulation

     $X_{true,sim,sfc}$     'True' surface profile level CO VMR profile used in retrieval simulation

**Appendix B**

In Section 3.1.3 we state that the surface profile level accounts for 55 % of $X_{sim,L3-L,sfc}$ departure from
$X_{sim,apr,sfc}$, vs only 25 % for $X_{sim,L3-W,sfc}$. These values are calculated using the surface level row of the $AK_{sfc}$
* $\Delta$ column from Table B1 (highlighted yellow) and the sum of that row (highlighted blue).

For $X_{sim,L3-L,sfc}$: $\frac{-0.0389}{-0.0701} = 0.55$.

For $X_{sim,L3-W,sfc}$: $\frac{-0.0018}{-0.0075} = 0.25$.

**Table B1** Case study information and full calculations for simulated $L3_L$ and $L3_W$ profile retrievals ($X_{sim,L3-L}$ and $X_{sim,L3-W}$ respectively) for 2013-08-18. Table includes $log_{10}$(VMRs) used in the simulation and values for $AK_{sfc}$ * $\Delta$ ($\Delta = X_{apr,sim} - X_{true,sim}$) in $log_{10}$ space, to illustrate the effect of interlevel correlation on the simulated VMR retrieval. Values that are underlined indicate the maxima (by magnitude) for that column.
Simulated profile values for levels 900-100 hPa are shown in brackets because AKs and calculations for
these levels are not discussed in the text and $AK_{900}$ through $AK_{100}$ are not shown.

| Profile level | Input profiles | | | | | | $Ak_{sfc}$ (== $A[1,i]$) | | $AK_{sfc}$ * $\Delta$ (log$_{10}$) | | Simulated profiles ($X_{sim}$) | | | |
| --- | --- | --- | --- | --- | --- | --- | --- | --- | --- | --- | --- | --- | --- | --- |
| | $X_{apr,sim}$ | | $X_{true,sim}$ | | $\Delta$ | | | | | | $X_{sim,L3-W}$ | | $X_{sim,L3-L}$ | |
| | VMR | Log$_{10}$ | VMR | Log$_{10}$ | VMR | Log$_{10}$ | L3$_W$ | L3$_L$ | L3$_W$ | L3$_L$ | Log$_{10}$ | VMR | Log$_{10}$ | VMR |
| Surface | 188.81 | 2.28 | 128.45 | 2.11 | -60.36 | -0.17 | 0.01 | 0.23 | -0.0018 | -0.0389 | 2.27 | 185.58 | 2.21 | 160.67 |
| 900 hPa | 148.88 | 2.17 | 119.53 | 2.08 | -29.34 | -0.10 | 0.03 | 0.20 | -0.0025 | -0.0191 | (2.15) | (141.49) | (2.09) | (124.02) |
| 800 hPa | 118.07 | 2.07 | 109.80 | 2.04 | -8.27 | -0.03 | 0.02 | 0.15 | -0.0008 | -0.0048 | (2.04) | (110.76) | (2.00) | (100.25) |
| 700 hPa | 102.72 | 2.01 | 97.64 | 1.99 | -5.08 | -0.02 | 0.03 | 0.14 | -0.0006 | -0.0030 | (1.98) | (95.37) | (1.95) | (88.72) |
| 600 hPa | 95.49 | 1.98 | 90.34 | 1.96 | -5.15 | -0.02 | 0.03 | 0.12 | -0.0007 | -0.0028 | (1.94) | (87.92) | (1.92) | (83.64) |
| 500 hPa | 92.00 | 1.96 | 87.10 | 1.94 | -4.91 | -0.02 | 0.03 | 0.08 | -0.0006 | -0.0020 | (1.93) | (84.57) | (1.92) | (82.57) |
| 400 hPa | 89.64 | 1.95 | 83.85 | 1.92 | -5.78 | -0.03 | 0.02 | 0.02 | -0.0005 | -0.0006 | (1.92) | (83.46) | (1.92) | (84.12) |
| 300 hPa | 84.76 | 1.93 | 81.42 | 1.91 | -3.34 | -0.02 | -0.01 | -0.07 | 0.0002 | 0.0013 | (1.91) | (81.73) | (1.92) | (83.91) |
| 200 hPa | 64.45 | 1.81 | 65.20 | 1.81 | 0.75 | 0.01 | -0.01 | -0.04 | 0.0000 | -0.0001 | (1.80) | (63.77) | (1.81) | (64.75) |
| 100 hPa | 26.51 | 1.42 | 28.71 | 1.46 | 2.20 | 0.03 | 0.00 | 0.00 | 0.0001 | -0.0000 | (1.42) | (26.41) | (1.42) | (26.52) |
| | | | | | | Rowsum → | 0.15 | 0.84 | -0.0075 | -0.0701 | ← Sum | | | |



**Author contributions**

IA and AW jointly conceived of and designed the study. IA performed data analysis; both authors examined and interpreted the results, and prepared the manuscript.

**Competing interests**

The authors declare that they have no conflict of interest.

**Acknowledgements**

The authors received funding from the Canadian Space Agency through the Earth System Science Data Analyses program (grant no. 16SUASMPTN), the Canadian National Science and Engineering Research Council through the Discovery Grants Program, and Saint Mary's University. We thank the MOPITT team and ECMWF for providing the data used in this study, and would also like to thank Dylan Jones (University of Toronto) and James Drummond (Dalhousie University) for their helpful discussions.

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





**Table 1 (a)** Surface classification of L3 data for the 1° x 1° gridbox containing Halifax (L3$_O$) for the period
2000-03-03 to 2017-03-05. Note that no retrievals are available for this L3 gridbox on 93 % of days in DJF
and 83 % of days in JJA. **(b)** Mean number of individual L2 retrievals with "land" and "water" surface index
that contribute to the L3$_L$ and L3$_W$ timeseries, respectively (the standard deviation is in brackets).

| | **(a)** | | | | **(b)** | |
|---|---|---|---|---|---|---|
| **Season** | **Water** | **Land** | **Mixed** | **Total** | **L3$_L$** | **L3$_W$** |
| DJF | 105 | 1 | 24 | 130 | 1.9 (±1.4) | 3.6 (±1.9) |
| JJA | 143 | 1 | 104 | 248 | 2.2 (±1.1) | 3.8 (±1.1) |

**Table 2** Case study information for simulated L3$_L$ and L3$_W$ profile retrievals (X$_{sim,L3-L}$ and X$_{sim,L3-W}$
respectively) for 2013-08-18. $\Delta$ = X$_{apr,sim}$ − X$_{true,sim}$. Values that are underlined indicate the maxima (by
magnitude) for that column. Simulated profile values for levels 900-100 hPa are shown in brackets because
AKs and calculations for these levels are not discussed in the text and AK$_{900}$ through AK$_{100}$ are not shown.

| Profile level | Input profiles | | | Ak$_{sfc}$ (== A[1,i]) | | Simulated profiles (X$_{sim}$) | |
|---|---|---|---|---|---|---|---|
| | X$_{apr,sim}$ | X$_{true,sim}$ | $\Delta$ | L3$_W$ | L3$_L$ | X$_{sim,L3-W}$ | X$_{sim,L3-L}$ |
| Surface | 188.81 | 128.45 | -60.36 | 0.01 | 0.23 | 185.58 | 160.67 |
| 900 hPa | 148.88 | 119.53 | -29.34 | 0.03 | 0.20 | (141.49) | (124.02) |
| 800 hPa | 118.07 | 109.80 | -8.27 | 0.02 | 0.15 | (110.76) | (100.25) |
| 700 hPa | 102.72 | 97.64 | -5.08 | 0.03 | 0.14 | (95.37) | (88.72) |
| 600 hPa | 95.49 | 90.34 | -5.15 | 0.03 | 0.12 | (87.92) | (83.64) |
| 500 hPa | 92.00 | 87.10 | -4.91 | 0.03 | 0.08 | (84.57) | (82.57) |
| 400 hPa | 89.64 | 83.85 | -5.78 | 0.02 | 0.02 | (83.46) | (84.12) |
| 300 hPa | 84.76 | 81.42 | -3.34 | -0.01 | -0.07 | (81.73) | (83.91) |
| 200 hPa | 64.45 | 65.20 | 0.75 | -0.01 | -0.04 | (63.77) | (64.75) |
| 100 hPa | 26.51 | 28.71 | 2.20 | 0.00 | 0.00 | (26.41) | (26.52) |
| | | | Rowsum → | 0.15 | 0.84 | | |





**Table 3** Results from OLS regression analysis of seasonal mean $L3_O$, $L3_L$ and $L3_W$ timeseries for selected profile levels in DJF and JJA. Units for TCO are mol cm$^{-2}$, all other levels are ppbv. m = gradient of OLS best-fit line; SE = standard error of gradient; p value = probability that the gradient is zero; % change y$^{-1}$= mean percentage change in retrieved CO per year, calculated from OLS regression model predicted values (pred) as follows:

$$\% \text{ change } y^{-1} = \left\{ \left[ \left( \frac{\text{Predicted}_{last}}{\text{Predicted}_{first}} \right) * 100 \right] - 100 \right\} / ny$$

where ny = number of years. Underlined and italicized rows indicate that the p-value associated with the gradient is > 0.1.

| Season | Level | Timeseries | m | SE | p value | % change y$^{-1}$ |
|---|---|---|---|---|---|---|
| **DJF** | | $L3_O$ | -1.97 | 0.31 | 0.000 | -0.99 |
| | Surface | $L3_W$ | -2.05 | 0.35 | 0.000 | -1.02 |
| n $L3_O$: 130 | | $L3_L$ | -1.59 | 0.81 | 0.075 | -0.85 |
| n $L3_W$: 127 | | $L3_O$ | -2.77 | 0.31 | 0.000 | -1.41 |
| n $L3_L$: 32 | 900hPa | $L3_W$ | -2.83 | 0.35 | 0.000 | -1.43 |
| | | $L3_L$ | -2.42 | 0.79 | 0.011 | -1.35 |
| | | $L3_O$ | -2.67 | 0.29 | 0.000 | -1.55 |
| | 800hPa | $L3_W$ | -2.64 | 0.33 | 0.000 | -1.53 |
| | | $L3_L$ | -2.44 | 0.65 | 0.003 | -1.54 |
| | | $L3_O$ | -0.95 | 0.44 | 0.047 | -0.74 |
| | 600hPa | $L3_W$ | -0.80 | 0.43 | 0.083 | -0.63 |
| | | $L3_L$ | -1.24 | 0.62 | 0.071 | -0.98 |
| | | $L3_O$ | 1.01 | 0.38 | 0.019 | 1.34 |
| | 300hPa | $L3_W$ | 1.09 | 0.34 | 0.006 | 1.44 |
| | | *$L3_L$* | *0.86* | *0.95* | *0.387* | *1.07* |
| | | $L3_O$ | -1.67E+16 | 4.40E+15 | 0.002 | -0.64 |
| | TCO | $L3_W$ | -1.58E+16 | 4.50E+15 | 0.003 | -0.61 |
| | | $L3_L$ | -1.84E+16 | 8.20E+15 | 0.045 | -0.74 |
| **JJA** | | $L3_O$ | -1.16 | 0.32 | 0.003 | -0.58 |
| | Surface | $L3_W$ | -0.51 | 0.28 | 0.092 | -0.26 |
| n $L3_O$: 248 | | $L3_L$ | -3.28 | 0.68 | 0.000 | -1.54 |
| n $L3_W$: 243 | | $L3_O$ | -1.85 | 0.46 | 0.001 | -1.08 |
| n $L3_L$: 106 | 900hPa | $L3_W$ | -1.64 | 0.42 | 0.001 | -0.96 |
| | | $L3_L$ | -3.34 | 0.78 | 0.001 | -1.83 |
| | | $L3_O$ | -1.80 | 0.51 | 0.003 | -1.25 |
| | 800hPa | $L3_W$ | -1.97 | 0.58 | 0.004 | -1.34 |
| | | $L3_L$ | -2.09 | 0.60 | 0.003 | -1.45 |
| | | *$L3_O$* | *-0.55* | *0.43* | *0.225* | *-0.51* |
| | 600hPa | $L3_W$ | -0.86 | 0.47 | 0.088 | -0.78 |
| | | *$L3_L$* | *0.03* | *0.45* | *0.942* | *0.03* |
| | | $L3_O$ | 2.19 | 0.44 | 0.000 | 2.80 |
| | 300hPa | $L3_W$ | 2.12 | 0.44 | 0.000 | 2.65 |
| | | $L3_L$ | 3.38 | 1.00 | 0.006 | 4.31 |
| | | *$L3_O$* | *-3.84E+15* | *6.30E+15* | *0.551* | *-0.16* |
| | TCO | *$L3_W$* | *-4.54E+15* | *5.90E+15* | *0.454* | *-0.19* |
| | | *$L3_L$* | *-2.12E+15* | *8.60E+15* | *0.810* | *-0.09* |




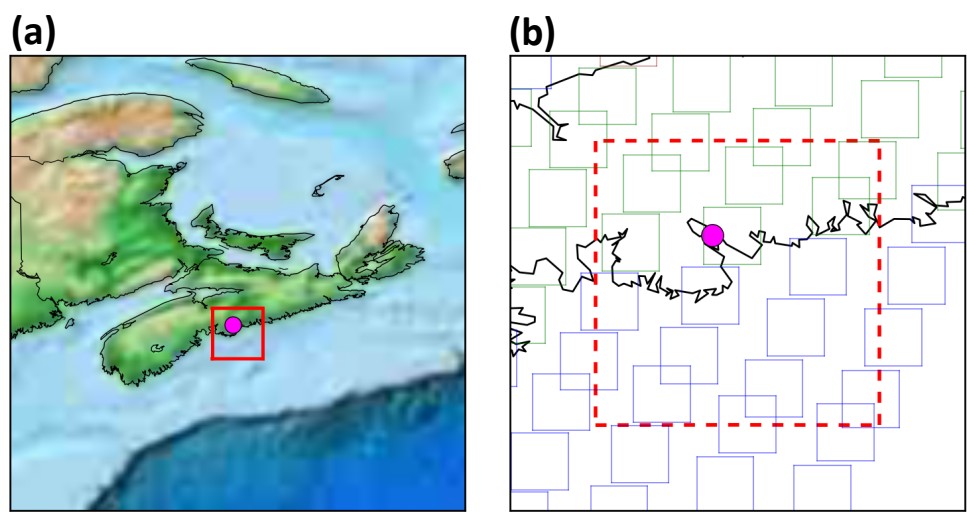

**Figure 1 (a)** Map of Nova scotia and the surrounding region in Atlantic Canada. Lighter colours on land indicate higher elevation. The pink icon shows the location of Halifax, the coastal city that we focus on in this study (see Section 2.1.3), and the red box shows the MOPITT L3 1° x 1° gridbox containing Halifax. **(b)**
Map zoomed to the MOPITT L3 gridbox containing Halifax (red dashed box), with the approximate location of individual L2 retrieval footprints shown (blue boxes = L2 surface index of water, green boxes = L2 surface index of land). L2 retrieval footprints with a midpoint that falls within the boundaries of the L3 gridbox will be averaged together to create the L3 data, according to certain rules – see Section 2.1.2 for full explanation.




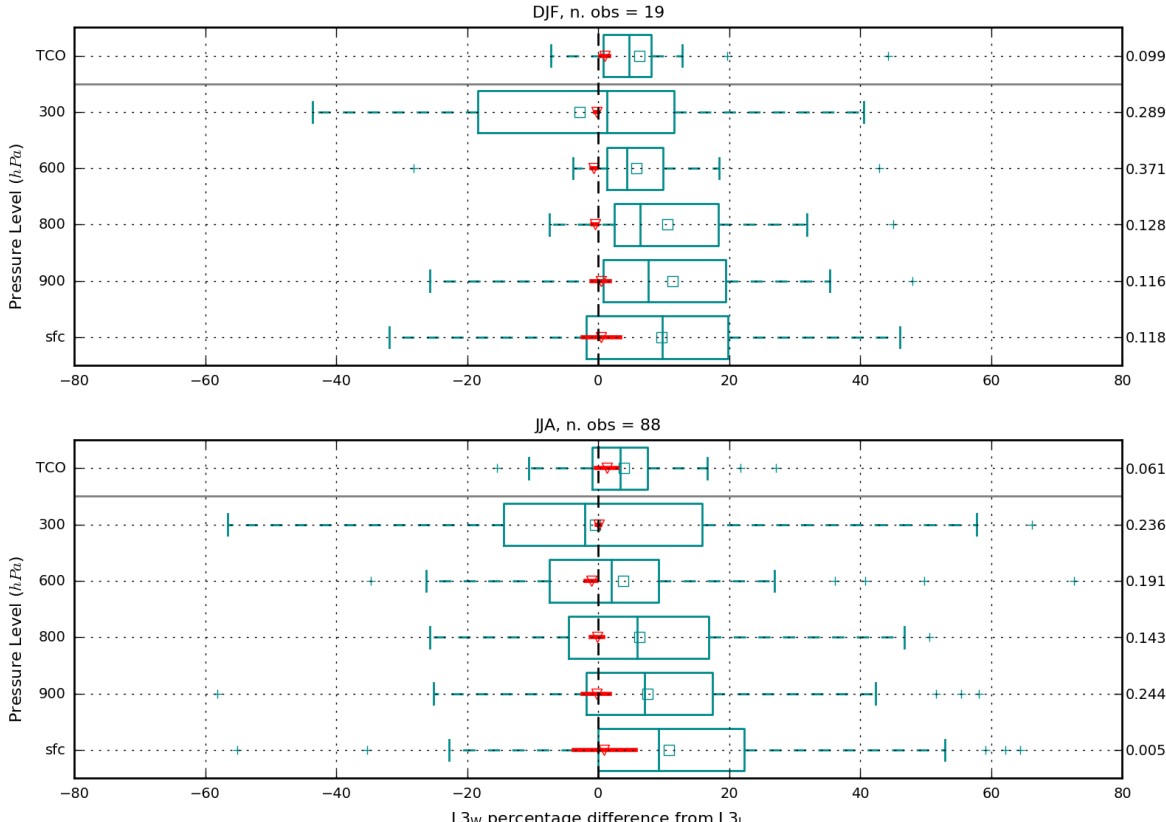

**Figure 2** Distribution of percentage difference[†] between temporally coincident retrieved VMRs (selected profile levels) and CO total column (TCO) from $L3_W$ and $L3_L$. Squares = mean differences, with the p value associated with each mean difference (from a 2-tailed Student's t test) given on the right-hand side y axis. Plus symbols = outliers[‡]. Red triangles = mean percentage difference between a priori values. Red lines = range of a priori difference values (where barely visible this means range is very small).

[†]Method for calculating percentage differences: $\left\{\left(\frac{L3_W}{L3_L}\right) * 100\right\} - 100$

[‡] Outliers defined as: above (below) percentile 75 (25) + (-) 1.5 * interquartile range





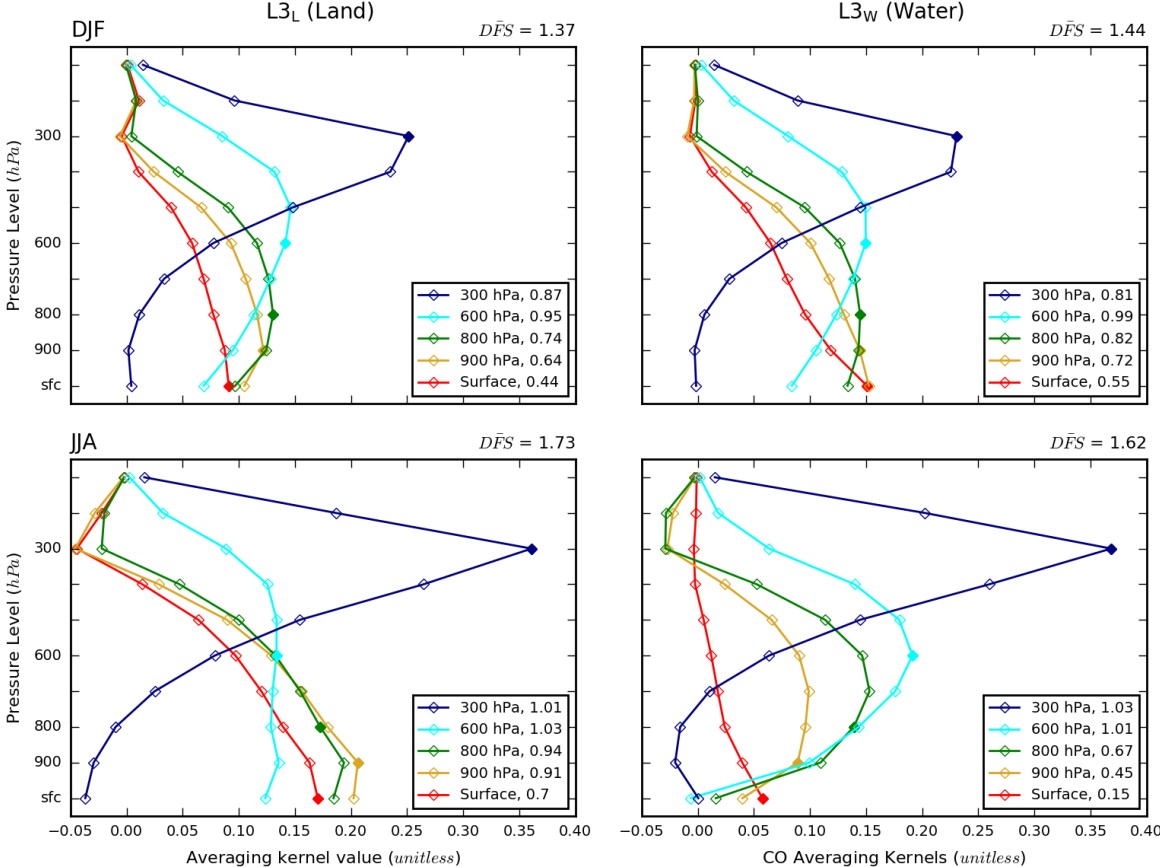


**Figure 3** Mean retrieval averaging kernel (AK) rows and DFS values for L3$_L$ ("Land", left column) and L3$_W$ ("Water", right column) in DJF (top row; n = 19) and JJA (bottom row; n = 88), selected profile levels only. Filled diamonds indicate diagonal value location for that AK. Numbers in legend indicate corresponding retrieval level of AK and show the mean rowsum for that AK level.






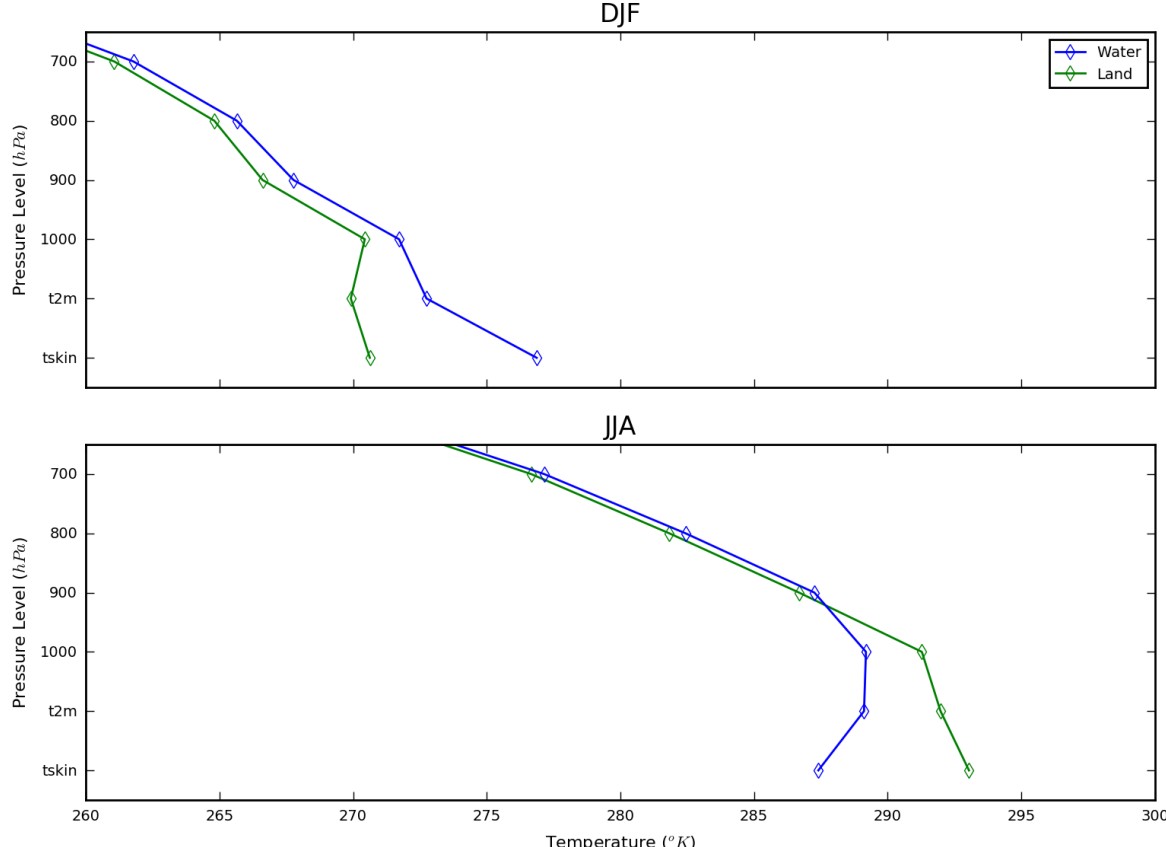

**Figure 4** Mean temperature profile from ERA-Interim for the closest all-land and all-water model gridboxes to Halifax. Data are the mean of the 12:00 UTC and 18:00 UTC timesteps (this corresponds to 08:00 and 14:00 local time, the closest ERA Interim timesteps to local MOPITT overpass time of ~10:30), for 2000-2017.



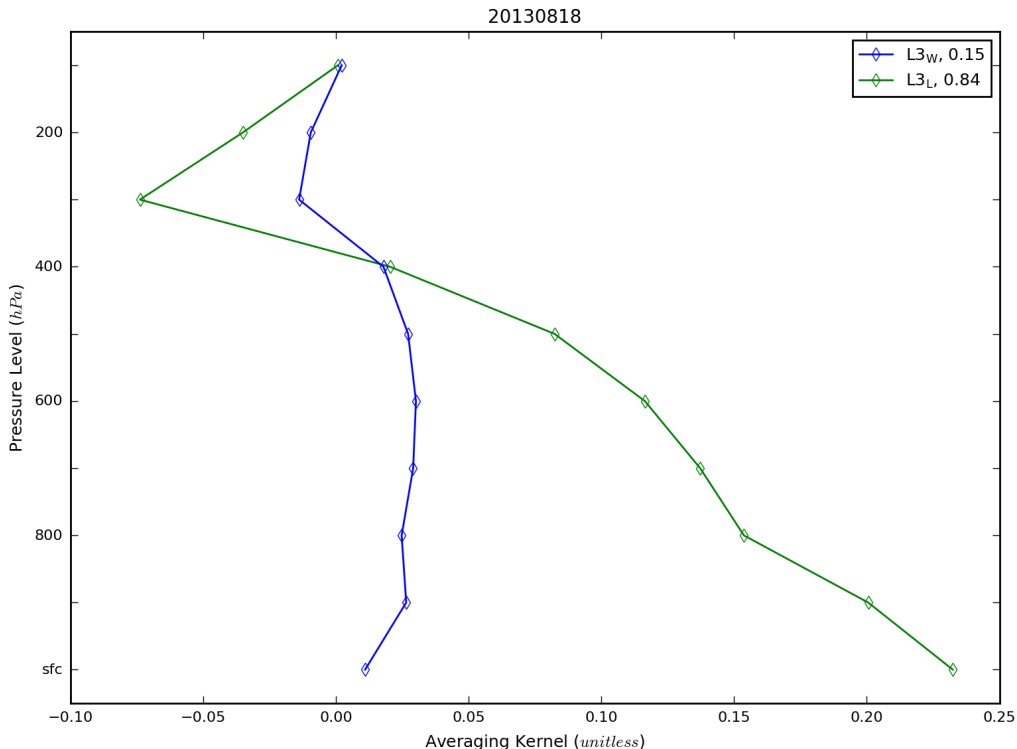

**Figure 5** Surface-level averaging kernels corresponding to L3$_L$ (green lines) and L3$_W$ (blue lines) for 2013-08-18 case study. Numbers in legend show the rowsum for that averaging kernel (AK$_{rowsum}$).



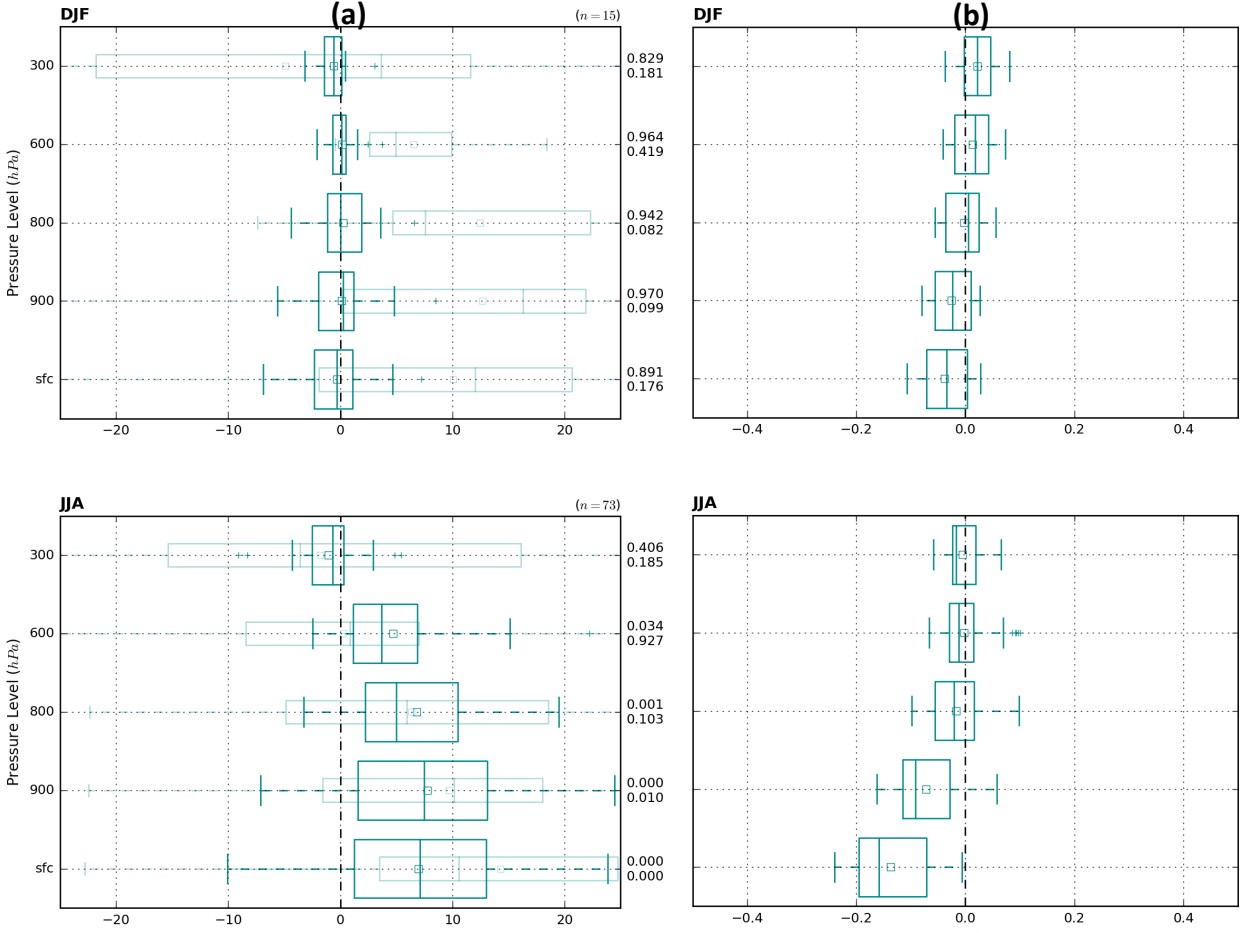


**Figure 6** (Next page) **(a)** Distribution of percentage difference[†] between simulated temporally coincident VMR retrievals in L3$_W$ (SIM$_{L3-W}$) and L3$_L$ (SIM$_{L3-L}$). Squares = mean differences. The p value associated with each mean difference (from a 2-tailed Student's t test) is given on the right-hand side y axis (top row = ΔSIM; bottom row = ΔRET). Plus symbols = outliers[‡]. Faint shading = corresponding ΔRET boxplots for comparison. Note that the sample size is different to Figure 2 owing to the CAMSRA data used as the X$_{true,sim}$ profile only covering a subset of the MOPITT years (2003-2016 vs 2000-2017 in Figure 2). The ΔRET boxplots overlaid cover this shortened period. **(b)** Distribution of the $\log_{10}X_{true,sim} - \log_{10}X_{apr,sim}$ values calculated during the simulation of retrieved profiles (see Equation 2) in DJF (top row) and JJA (bottom row).

[†]Method for calculating percentage differences: $\left\{\left(\frac{L3_W}{L3_L}\right) * 100\right\} - 100$

[‡] Outliers defined as: above (below) percentile 75 (25) + (-) 1.5 * interquartile range





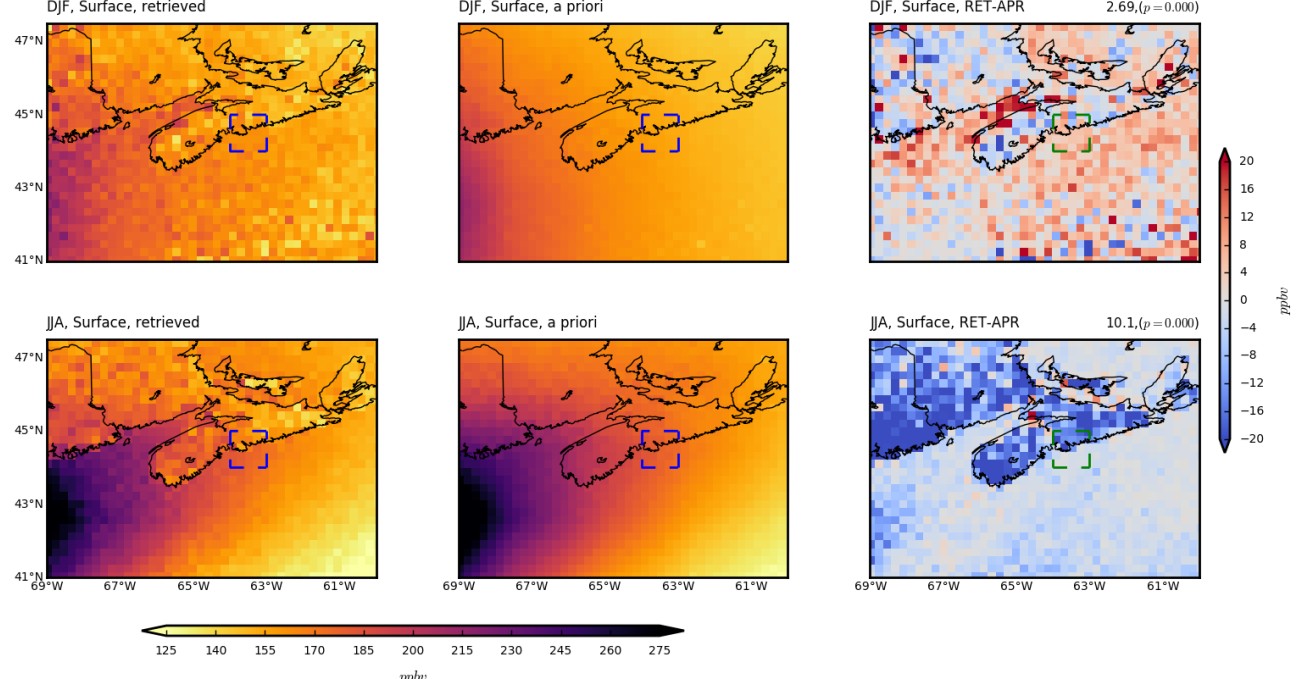

**Figure 7** Seasonal median L2[†] retrieved VMR (left column), a priori VMR (centre column), and RET-APR (right column) at the surface profile level in DJF (top row) and JJA (bottom row). Values to the right above RET-APR plots = $\overline{(\text{L2 retrievals over water})}$ − $\overline{(\text{L2 retrievals over land})}$ for plotted area (data were first binned according to L2 surface index); numbers in brackets correspond to significance of mean difference using a 2-tailed Student's t test. Blue or green dashed square = outline of L3 grid box that contains Halifax.

[†]These maps were created from L2 data that were interpolated to a regular 0.25º x 0.25º grid for ease of plotting.





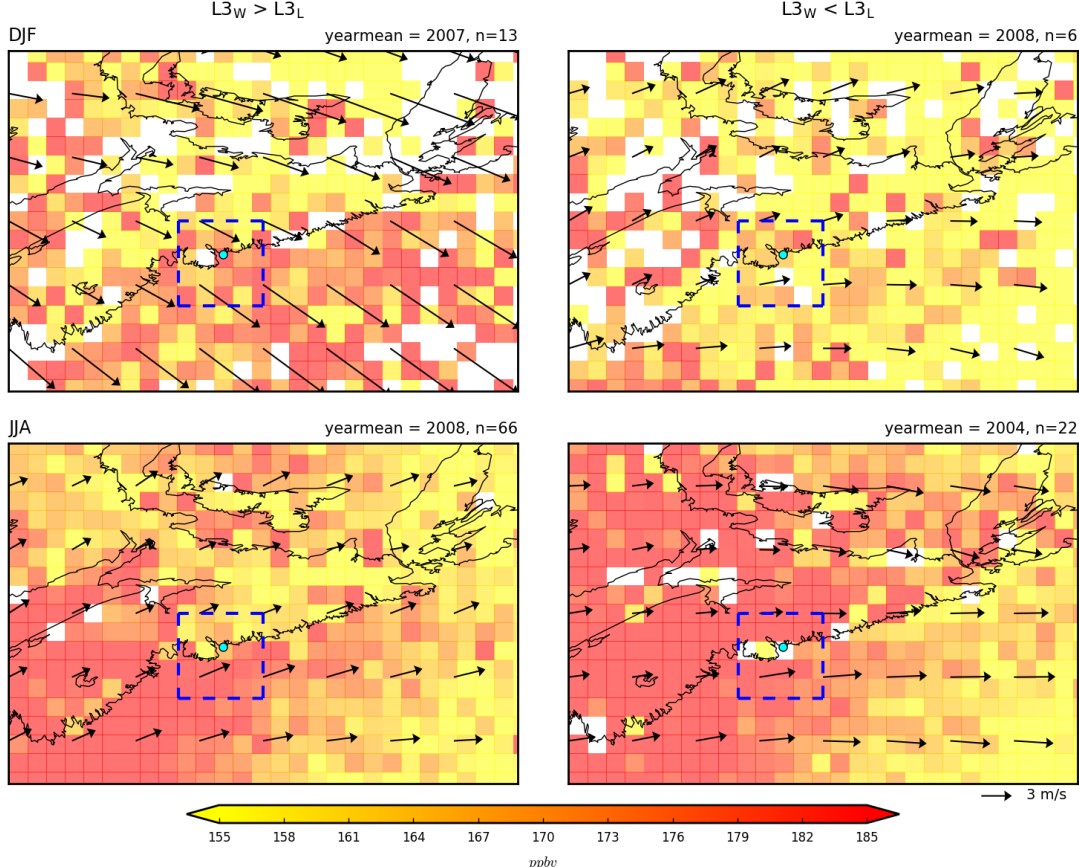


**Figure 8** Mean ERA-Interim 10-metre winds (vectors) and MOPITT L2[†] VMR at the surface (shading) for days when retrieved surface level VMRs in $L3_W$ are greater than in $L3_L$ ($L3_W > L3_L$) and days when they are less ($L3_W < L3_L$). Top row = DJF; bottom row = JJA. "Yearmean" corresponds to the mean year in which retrievals from the respective samples took place (study period spans 2000-2017).

[†]These maps were created from L2 data that were interpolated to a regular 0.25° x 0.25° grid for ease of plotting.




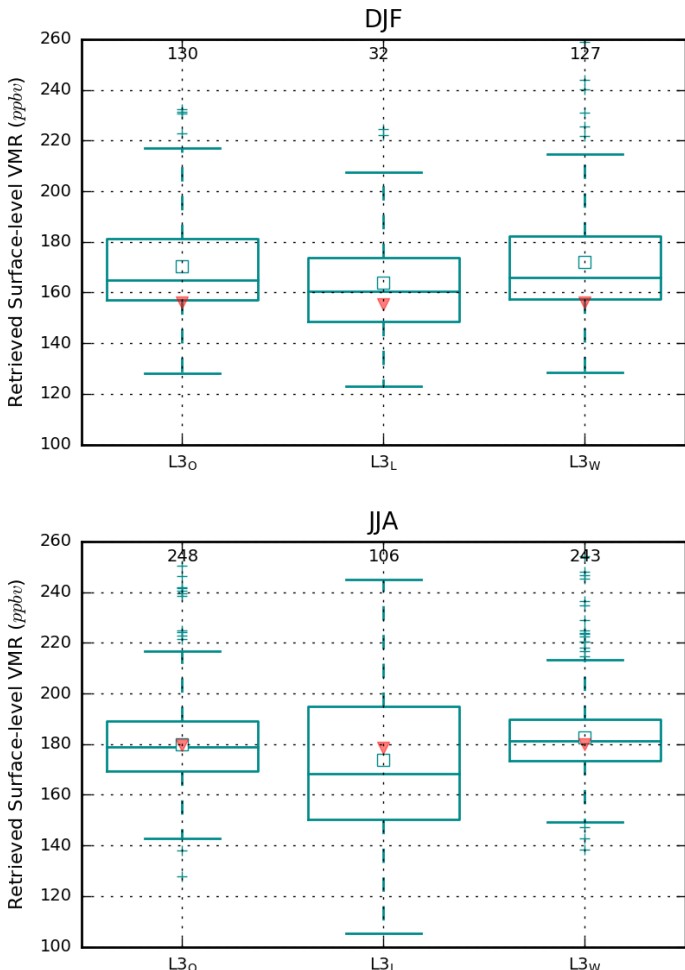

**Figure 9** Seasonal distribution of retrieved surface level VMR values from $L3_O$, $L3_L$ and $L3_W$. Squares = mean values; Red triangles = corresponding mean a priori values. Sample sizes are given below the top x axis.




**Figure 10** (next page) OLS regression best-fit lines calculated from seasonal mean timeseries for DJF (top) and JJA (bottom) of L3$_O$ (black), L3$_L$ (green) and L3$_W$ (blue) retrieved surface-level VMR. The daily observations corresponding to each seasonal mean value are represented by colour-coded boxplots each year (horizontal lines in middle of boxes = seasonal median value; filled squares = seasonal mean value). The

dashed red line is the mean of the corresponding seasonal mean L3$_O$, L3$_L$ and L3$_W$ a priori data. Colour-coded values below the top x axis correspond to the number of observations each season. Values in the legend are the value, standard error, 95% confidence limits and probability of zero value of the gradient parameter, respectively.