# Peer review of "Impact of land-water sensitivity contrast on MOPITT retrievals and trends over a coastal city"

_Atmospheric Measurement Techniques, 2019_

## Referee Comment (RC1) · Anonymous Referee #1 · 25 Jan 2020

This study examines how MOPITT version 7 TIR-NIR retrievals compare over land versus water over and around a single coastal city – Halifax, Canada. The authors examine the L3 product as well as the L2 product averaged separately for land and water over the same grid box. They generally find that retrievals over water retrievals are higher than over land. Most of this difference can be accounted for by differences in averaging kernels. While this paper has numerous strengths (nearly flawless grammar, and very detailed), I think a step back is needed to understand the overarching goal of this study or central question to be answered. This may involve a scope change from e.g., a very local to a global scale, examining additional MOPITT products, and/or including additional measurement results or results from a model the authors run. While

[Figure]

I cannot recommend publication in its current form, I encourage resubmission when the comments below are addressed.

General (Some of these general comments may supersede specific comments below).

G1: The paper is quite detail oriented, but at times there is so much repetition and additional words and phrases that it makes it difficult to read. I think the length could be cut down by as much as 50%. Part of this could be through wording changes, part of it by reorganization, and part of it fewer numerical details that could be succinctly summarized in Tables. Aim for describing things in context rather than the large number of uncommon symbols and abbreviations.

G2: One of the major conclusions of this paper is caution in examining retrievals around coastlines throughout the world. This conclusion is not supported by this paper. Instead this paper focuses on only one coastal grid box of hundreds across the globe. All references that imply potentially large differences elsewhere need to be removed or a study needs to be undertaken to look at all coastlines.

G3: MOPITT version 8 has been available for a year, but version 7 was used. While there is nothing wrong with using an older version, V8 should at least be mentioned with a note that the difference may or may not remain in V8. Further, when comparing land and water soundings the TIR product is probably a better choice than TIR-NIR as NIR bands are only used over land. Retrievals from other seasons should also be considered for completeness, even if not examined in as much detail.

G4: A major conclusion seems to be that averaging kernels need to be accounted for. When they are, the differences in retrieved amounts over land compared to water decrease significantly. The need to account for averaging kernels has already been known in the remote sensing community for decades.

G5: A recommendation in this paper is that all soundings over water should be discarded. This would represent a significant loss of information. The argument is that

individual land soundings have greater information content (which is not surprising as land soundings use both TIR and NIR, when water soundings can only use TIR). If this argument were extrapolated further, one might say to only use soundings with degrees of freedom of signal (DFS) of say 1.8 or larger. While this would also maximize information content of individual soundings, it would likely decrease the information content from the MOPITT record of the Earth system as a whole. An atmospheric model is needed to substantiate the advice of discarding all retrievals over water.

G6: P-values are used throughout, but their implications often need more consideration. A small p-value may indicate a statistical difference in the mean, but does not say why the difference appeared. In this study it seems the difference can mostly be accounted for by differences in averaging kernels (which are already known to be important). When 93% of days do not have the right data, it makes it difficult to draw conclusions.

Specific comments

P2L41: CO is the only target gas from MOPITT (CH4 cannot and will not be retrieved from the observations).

P2L43: A reference is needed for CO lifetime.

P2L43: Volatile organic compounds contribute (indirectly) to about half of CO in Earth's atmosphere.

P2L63: What is meant by "information content" here? DFS? Shannon information content? (They are related, so maybe this is meant to be generic?)

P5L130: Include a reference to the a priori.

P5L133: Watch your usage of "layers" versus "levels" here and throughout. Retrievals are on layers, but are reported on levels for MOPITT. Note there are only 8 layers from 900 to 100 hPa (when surface pressure is greater than 900 hPa). The uppermost layer is 100 to 50 hPa.
P5L141: Equation 1 is missing the error term.

P6L163: "Is generally advised against" – please provide a reference.

P6L165: These supposed guidelines need to be clarified. While such restrictions may increase the average information content of individual soundings, the restrictions may decrease the information content of the system as a whole.

P6L173: Maybe this many significant figures are what are reported in the census, but it seems like too many. What if someone moves to Halifax?

P6L174: Briefly describe the pollutants.

P6L177: It appears that 2 more years of MOPITT V7 data are now available.

P7L205: A brief explanation about the 4 MOPITT pixels is needed here.

P8L223: I dislike the use of "true" for model values. Please modify throughout.

P13: Watch significant figures throughout.

Table 3: Is the purpose of p-values here to show that CO levels are changing in the MOPITT record?

Briefly describe why OLS was used.

Figure 6: "Next page" (?)

Figure 8: Are the bounding boxes shown correct for a 1 degree box?

---

## Referee Comment (RC2) · Anonymous Referee #2 · 26 Jan 2020

Ashpole and Wiacek analyze a MOPITT CO Level 3 (L3) pixel over one location – Halifax, Canada – and compare it to the Level 2 (L2) retrievals within the same 1 degree pixel. They use this coastal location to highlight instrument sensitivity differences between water and land that impact near-surface and profile analysis with the joint TIR-NIR product. The influence of different surface-types (land or water) on CO retrievals and the resulting trends is investigated. The authors find that sensitivity differences account for retrieval differences in JJA, but a CO gradient is likely the reason for differences in DJF. While the MOPITT team already provide recommendations to maximize information content for a studied region, Ashpole and Wiacek demonstrate the practical implications of retrieval differences. The study suggests L2 profile data over land is

more appropriate than L3 profile for the small region around the coastal city of Halifax, particularly for the lower troposphere. The authors present a valuable supplementary guide for users of the MOPITT product.

Overall the analysis shows attention to detail and generally good statistical practices. The manuscript is well written and language is clear with very few technical corrections. I have some recommendations for reducing verbosity. Additionally, I recommend the following major comments be addressed before the manuscript is considered for publication.

**Major comments:**

1. Section 3.2: In my opinion, the most logically inconsistent part of the manuscript is the comparison of all $L3_O$ pixels with $L3_L$ and $L3_W$. The surface pixel type (land, water or mixed) is given in the MOPITT level 3 product, and users should use information to filter data over such locations. I think the strength of this analysis would be to show that the improvement in information content if you use the L2 land data over coastal locations. Also - is there a difference between using L2 water and the L3 water retrievals, i.e. the extra days you gain from the "mixed" L3 pixel when creating $L3_W$. The authors need to clearly justify why they are comparing with the combination of L3 pixel-types, or alternatively compare against the water-only pixels from Level 3.

2. Trend analysis: The MOPITT profile layers are known to have drift (Deeter et al., 2017, doi:10.5194/amt-2017-71). Surface drift is about -0.7% per year, 800 hPa is -1% per year. The UT levels have positive drift - both the 400 and 200 hPa levels drift at greater than 1% per year. Drift should be corrected before trend analysis is performed. I did not see this mentioned in the manuscript.

3. Ordinary Least Squares analysis is highly susceptible to outliers and end points. End points may in particular be impacting the slope calculations for $L3_L$ data in JJA (Figure 10, bottom panel). Instead, I recommend that the trend analysis be performed with weighted least squares (WLS), weighted by the standard deviation, which is much

less susceptible to outliers. Additionally, the seasons that only have one day per 3 months can be de-weighted by using sufficiently large standard deviation.

4. Please mention somewhere (for example on P6, L189) why the analysis is statistically valid, even though only 7% of the potential DJF days and 17% of the potential JJA days have measured CO.

5. I am uncomfortable with the authors extending this analysis to other regions (e.g. P2, L35; P20, L605 to L607). Before extending these results to other coastal locations, other types of locations need to be analyzed (tropical, temperate). Please add a comment that recommends similar analysis be performed for other coastal sites before using MOPITT surface CO.

6. There was a change in MOPITT processing after the cooler failure in May 2001. Although a homogenized record is attempted, there remains a small step-change in data. I suggest to use data only from the latter part of the record starting August 2001, especially for the trend analysis where step changes could have large impact.

7. The authors have missed an opportunity to quantify the improvement in retrievals due to the inclusions of NIR between $L3_L$ and $L3_W$. In the JJA comparisons, when the CO gradients and a priori suggest there are no expected differences in the CO.

8. Figure 6 and associated discussion. I suggest to split the $logX_{true}$ - $logX_{ap}$ into different average profiles for land and water because Figure 7 suggests that in JJA both difference would be negative while for DJF, the land difference would be negative but the water would be positive. This would help support the overall argument that DJF sees real CO gradients, while JJA is impacted by sensitivity differences.

**Minor comments:**

P1, L9: Add latitude and longitude to location in abstract.

P1, L18 L30 and elsewhere in manuscript: Change "surface profile level" to just "surface level" - otherwise it should be "surface level of the profile".

[Figure]

P2, L41: Remove "primary" from "The primary target" because MOPITT only retrieves CO.

P2, L46: Add in a sentence about the secondary production of CO from VOCs.

P2, L48: "...since launch in December, 1999".

P4, L114 to 116: It is important to note here that the improved sensitivity occurs only over land for the joint TIR-NIR product. Also, largest difference would consequently be expected between land and water with the joint product, so the study presented in this manuscript is expected to show an upper bound on the differences in retrievals between land-types.

P4, L127: The simulation climatology resolution is $1.9° \times 2.5°$, not $1° \times 1°$.

P6, L172: Add degree symbols to the latitude and longitude locations.

P9, L274: I think it should be "MT and UT".

P10, L299: I was confused by "the widest part of each AK" - Did you mean in the X-direction or the Y-direction? Please clarify.

P11, L328: Do you mean differences in the MT?

P12, L346 to L347: Reword to be a little clearer, e.g. "The $T_{skin}$ difference approaching $20°$ K between JJA $L3_L$ and DJF $L3_W$ likely..."

P12, L370: Reword to clarify, e.g. "Available CAMSRA data covers 2003-2016, so a subset of the available MOPITT data (2000-2017) are considered in simulation experiments."

P13, L377: Remove the "respectively" explanation in brackets because there is only one number.

P13, L378: "...sensitivity to $X_{true,sim}$ in $L3_L$ than $L3_W$:..."

P15, L463: Change all-caps TRUE.

P16, L472 to L473: This is the first time the data is separated in this way ($L3_L > L3_W$ and $L3_W > L3_L$). Mention why this is necessary. Why is it not separated this way in the other sections - for example Figure 2 shows a mean response.

P16, L491 to L496: This description needs a little clarification. Could it be explained that the density of days where $L3_W < L3_L$ is higher at the beginning of the record, while the density of days where $L3_W > L3_L$ is higher at the end of the record for JJA. In DJF, the density of $L3_W > L3_L$ is consistent across the record.

P17, Section 3.2.1: Discuss/show p-values for all comparisons between means.

P18, L549: Consider discussing as "gradients for $L3_W$" etc. rather than introducing more acronyms. This would also help when reading Table 3 as the new acronyms are not used there.

P20, L604: "... before analysis of profile values in order to...".

P20, L605: "In particular, when investigating coastal cities, it is advised that retrievals over land..."

Table 3: Add units for m and SE.

Figure 4: Add error bars for standard deviation.

Figure 8: "Yearmean" is an unusual metric. I suggest "Center year".

Figure 9: What are the error bars?

**Suggestions to reduce verbosity:**

P5, L155: Remove end of sentence from "in the case of" onwards - redundant.

P8, L227-L228: Remove "For simulated..." onwards - redundant.

P12, L348 to L361: This is a repeat of earlier and later information. Consider whether any or all of it is really needed here.

P12, L366 to L369: Remove sentence starting "For each $L3_L$-$L3_W$..." because repeats information from the methods section and is not needed here. Can just start sentence on line 369 with "Recall that any differences..."

P13, L382 to L383: Ending of sentence starting "as would be expected from..." can be removed because it is a repeat of L378 to L380.

P13, L384 to L391: Unsure that this adds any important points to the discussion, consider removing or shortening.

P16, L496: "As we show in Section 3.2.2,..." - Is this really needed here, or could you leave it till Section 3.2.2.

P17, L510 to L511: Remove end of sentence beginning "primarily because..." - repeat of information in the beginning of the sentence.

P17, L511 to L514: Remove "Thus, there is..." - it is unnecessary.

P17, L524 to L526: Consider removing "(mean surface level..." to "...248 for $L3_O$)".

P19, L581 to L587: There is no significant difference between trends at 300 hPa , so there is no need to explain why they might be different. This paragraph just needs to mention the main point that trends at 300 hPa are positive.

Table 2: $AK_{sfc}$ are in Figure 5 so can be removed. Consider adding other measures from Table 2 to Figure 5 and removing Table 2.

---

## Author Comment (AC1) · 24 Apr 2020

We thank Anonymous Referee #1 for their time in reviewing this manuscript. Our responses are in the attached supplement.

Please also note the supplement to this comment:
https://www.atmos-meas-tech-discuss.net/amt-2019-430/amt-2019-430-AC1-supplement.pdf
* * *

---

## Author Comment (AC2) · 24 Apr 2020

We sincerely thank the reviewers for their very helpful feedback on this paper. We address all their comments and suggestions below.

Reviewer comments are in black.

Our responses are in blue.

Where practical/necessary, we provide a screenshot of the track-changes document to show the changes that we have made (in outlined boxes). In these, text that is removed is  and coloured red, while new text is underlined and coloured blue.

Ian Ashpole (on behalf of both authors).

This study examines how MOPITT version 7 TIR-NIR retrievals compare over land versus water over and around a single coastal city – Halifax, Canada. The authors examine the L3 product as well as the L2 product averaged separately for land and water over the same grid box. They generally find that retrievals over water retrievals are higher than over land. Most of this difference can be accounted for by differences in averaging kernels. While this paper has numerous strengths (nearly flawless grammar, and very detailed), I think a step back is needed to understand the overarching goal of this study or central question to be answered. This may involve a scope change from e.g., a very local to a global scale, examining additional MOPITT products, and/or including additional measurement results or results from a model the authors run. While I cannot recommend publication in its current form, I encourage resubmission when the comments below are addressed.

**General comments (Some of these general comments may supersede specific comments below).**

G1: The paper is quite detail oriented, but at times there is so much repetition and additional words and phrases that it makes it difficult to read. I think the length could be cut down by as much as 50%. Part of this could be through wording changes, part of it by reorganization, and part of it fewer numerical details that could be succinctly summarized in Tables. Aim for describing things in context rather than the large number of uncommon symbols and abbreviations.

Thank you for the feedback. We have done a thorough edit which entailed removing a lot of the repetition and 'over-explaining' and aiming to be much more succinct. We do not repeat values or numbers that are clear in figures or tables that we are discussing, apart from when it is absolutely essential to emphasise the point. We have also removed some of the acronyms to make certain sections easier to follow (especially Section 3.1.3). We sincerely hope that this makes the article easier to read.

G2: One of the major conclusions of this paper is caution in examining retrievals around coastlines throughout the world. This conclusion is not supported by this paper. Instead this paper focuses on only one coastal grid box of hundreds across the globe. All references that imply potentially large differences elsewhere need to be removed or a study needs to be undertaken to look at all coastlines.

We apologise for making these claims. There are statements along these lines on 3 separate occasions in the original submission (in the abstract, introduction, and conclusion), and we have toned the wording down in all cases.

Abstract:

> the findings also apply to MAM and SON.  The results that we report here suggest that similar analyses be performed for other coastal cities before using MOPITT surface CO.

Introduction:

> likely to be targets for analyses of temporal trends in air quality indicators. The results that we report here suggest that similar analyses be performed for other coastal cities before using MOPITT surface CO .

Conclusion:

> in order to maximize the information content of L3 data in coastal gridboxes. Although our study has only focused on the city of Halifax, the results suggest that similar studies be performed for other coastal L3 gridboxes before using MOPITT surface CO,  since these contain 6 of the top 10 and 43 of the top 100 agglomerations by population and are therefore likely targets for analysis of temporal changes in air pollution indicators such as CO, especially near to the surface. The

G3: MOPITT version 8 has been available for a year, but version 7 was used. While there is nothing wrong with using an older version, V8 should at least be mentioned with a note that the difference may or may not remain in V8. Further, when comparing land and water soundings the TIR product is probably a better choice than TIR-NIR as NIR bands are only used over land. Retrievals from other seasons should also be considered for completeness, even if not examined in as much detail.

There are 3 components to this comment, which we address separately:

MOPITT v8: Unfortunately, this product was released after our analysis was completed. Thank you for accepting that work with the previous product is still valid. We have included references to v8 in a couple of locations where it is relevant, e.g, in Section 2.1.1:

> release of new product versions, with enhanced validation statistics against in situ CO observations. The work presented in this manuscript is based on MOPITT Version 7 (V7) products (Deeter et al., 2017). We analyse both L2 and L3 products (as outlined below). It should be noted that MOPITT Version 8 products have been released very recently, incorporating an improved radiance bias correction method to address a documented drift and geographical variability in retrieval bias compared to in-situ measurements (Deeter et al., 2019). It remains to be seen whether the impacts of land-water retrieval sensitivity contrasts documented in this study remain in this newest product version.

TIR-only product: Our justification for choosing the joint TIR-NIR product is the greater LT sensitivity it offers compared to TIR-only, as stated by several references that we cite. The fact that NIR bands are only used over land contributes to the land-water LT sensitivity contrast that then creates problems when L2 land and water retrievals are averaged together, as is the case for coastal L3 gridboxes. This is the main point we make in the paper. We have performed a supplementary analysis using the TIR-only product and find that the land-water contrast remains, albeit more weakly. This supports our claims made in Section 3.1.2 that a key source of difference in retrieval sensitivity in the lower troposphere is thermal contrast differences over land and water, as supported by previous studies (e.g. Deeter et al. 2007). We now mention the TIR-only product in our Data and Methods section (Section 2.1.1), and that we expect our analysis with the joint TIR-NIR product to represent an upper-limit on the land-water retrieval differences owing to the inclusion of the NIR data (see screenshot from track-changes document below). We include the analysis with TIR-only data in our Supplementary Material (SM1), and also refer to it in Section 3.1.2 when explaining the cause of the land-water sensitivity contrast, pointing out that while thermal contrast in the LT plays an important role, the NIR contribution being limited to retrievals over land will also have an effect (although we also note that a sensitivity contrast exists even in the TIR-only data).
* * *
lower troposphere (LT), to which TIR radiances are typically less sensitive (Pan et al., 1995, 1998). NIR radiances can, however, only be exploited in daytime scenes over land. Our results are based on analysis of the TIR-NIR combined product, owing to its greater sensitivity to LT CO compared to the TIR- and NIR-only products which are also available (e.g. Deeter et al., 2017). Owing to the increased LT sensitivity from NIR radiances being limited to retrievals over land, we expect that the results presented here show an upper bound on the retrieval differences between surface types within our coastal L3 gridbox of focus, and the consequent effects on sample statistics and temporal trends that we outline. Differences are still found in the

TIR-only product, however, and we outline these in Supplementary Material 1. We restrict our analysis to
* * *
Retrievals from other seasons: Thank you for raising this. We have now included a sub-section that briefly discusses the MAM and SON seasons at the end of the results and discussion section ("Section 3.3. Consideration of MAM and SON"), which ultimately demonstrates that the findings for DJF and JJA hold for these seasons too. We show mean averaging kernels to demonstrate that in MAM there are strong sensitivity differences in the lower troposphere (similar to JJA), while in SON the sensitivity contrast is much smaller (similar to DJF). We then demonstrate how the temporal changes detected in surface level VMRs in the different datasets studied are affected by these differences, with the datasets based on retrievals over water (what we call L3W and the original L3 dataset) underestimating the change in MAM (as in JJA), whereas there is very little difference between datasets in SON (like in DJF).

G4: A major conclusion seems to be that averaging kernels need to be accounted for. When they are, the differences in retrieved amounts over land compared to water decrease significantly. The need to account for averaging kernels has already been known in the remote sensing community for decades.

This is true. In our paper we demonstrate a practical consequence of this, for a widely-used data product. The point that we make in our paper is, essentially, that the information content of 1º x 1º Level 3 gridded MOPITT products in a coastal location is significantly affected by the fact that the averaging kernels are different for the retrievals over land and water from which it is made. We show that this has significant consequences for the results of temporal trend analysis with the data – the type of analysis which the MOPITT dataset is well-suited to owing to its long timespan. We therefore strongly believe that this is worthy of communication to the scientific community.

G5: A recommendation in this paper is that all soundings over water should be discarded. This would represent a significant loss of information. The argument is that individual land soundings have greater information content (which is not surprising as land soundings use both TIR and NIR, when water soundings can only use TIR). If this argument were extrapolated further, one might say to only use soundings with degrees of freedom of signal (DFS) of say 1.8 or larger. While this would also maximize information content of individual soundings, it would likely decrease the information content from the MOPITT record of the Earth system as a whole. An atmospheric model is needed to substantiate the advice of discarding all retrievals over water.

The point that discarding retrievals = a loss of information content about the Earth system as a whole (which we interpret as meaning e.g. a loss of spatial and/or temporal coverage) is a fair one. We have now made sure to mention this when talking about different guidelines to filter the MOPITT product, as it is a practical consequence that is important to bear in mind (see screenshot below). However, we are not alone in recommending that certain retrievals are discarded (we cite multiple references to the MOPITT team making such recommendations), and we feel that our recommendations are simply a logical extension of this. On the recommendation that additional data (i.e. from an atmospheric model) are needed to substantiate the advice of discarding all retrievals over water: we agree that a study about the effects of discarding certain retrievals on overall information content about the Earth system as a whole would be highly interesting and valuable. However, this is beyond the scope of the work that we present, which we hope is an acceptable contribution to the scientific literature in itself.

> 190 Unfortunately, such filtering does lead to an overall loss of available retrievals for analysis, reducing the effective temporal and spatial coverage of the data.
>
> **2.1.3. Study area, time period, and MOPITT data processing in this study**

G6: P-values are used throughout, but their implications often need more consideration. A small p-value may indicate a statistical difference in the mean, but does not say why the difference appeared. In this study it seems the difference can mostly be accounted for by differences in averaging kernels (which are already known to be important). When 93% of days do not have the right data, it makes it difficult to draw conclusions.

We think that there are two issues raised by this comment, which we address separately:

The use of p-values: We use p-values to point out where mean differences are statistically significant; or where trends identified in regression lines are either significantly different from zero, or significantly different from one-another. Where p-values are not significant, we do not say much about the data, since we can't be sure that differences aren't due to chance/sampling. It is our understanding that use of p-values in this way is common practise? We do not use p-values to say *why* differences appeared – we explain possible causes for the significant differences once they have been identified (i.e. linking significant differences in retrieved surface level VMRs over land and water to averaging kernel differences).

The issue of missing data: the high proportion of missing data, especially in DJF, is frustrating, but also unavoidable when using these data. It certainly makes it difficult to draw conclusions about the time period covered *as a whole*, but it does not preclude the making of conclusions about the temporal subset of data being studied – especially where differences are found to be statistically significant. We now make sure to point out that due to the high proportion of missing data, our conclusions should not be taken as being representative of the time period as a whole. We do this in the Data and Methods section (2.1.3) and also re-iterate it in Section 3.2.2 when discussing the temporal trends we identify.

e.g. from Section 2.1.3:

> that were made during daytime hours. This yields a timeseries with one observation per day, when retrieval data were available within this gridbox. There are no retrievals available on 91 % of all days in DJF and 83 % of all days in JJA for the period covered. This is a result of both 1) MOPITT's polar orbit limiting temporal resolution to ~3 days over most of the globe; and 2) on days when the satellite's swath does encompass Halifax, retrievals either not being made due to cloud coverage, or discarded due to data quality issues. While this does not prevent a meaningful comparison of available retrievals, it does mean that caution is needed when using them to draw conclusions about the time period covered as a whole, which is something that we do not attempt to do.

**Specific comments**

P2L41: CO is the only target gas from MOPITT (CH4 cannot and will not be retrieved from the observations).

We have removed the mention of "primary target gas":

> the composition of the Earth's atmosphere from space. The  target gas for MOPITT is carbon monoxide (CO), whichemitted from a range of anthropogenic (e.g. fossil fuel use) and natural (e.g.

P2L43: A reference is needed for CO lifetime.

We have included a reference to Duncan et al. 2007 (JGR: "Global budget of CO, 1988–1997: Source estimates and validation with a global model")

> monoxide (CO), whichemitted from a range of anthropogenic (e.g. fossil fuel use) and natural (e.g. wildfires) sources, produced via the oxidation of methane and other volatile organic compounds, and has an atmospheric lifetime of weeks to months depending on season and location (e.g. Duncan et al., 2007).

P2L43: Volatile organic compounds contribute (indirectly) to about half of CO in Earth's atmosphere.

We now include VOCs in the list of CO sources mentioned (see screenshot above).

P2L63: What is meant by "information content" here? DFS? Shannon information content? (They are related, so maybe this is meant to be generic?)

We mean DFS. Since information content is discussed in detail in the next paragraph, we do not feel that further clarification is needed at this stage.

P5L130: Include a reference to the a priori.

We are not entirely sure what the reviewer is referring to here. The source of the a priori is mentioned in the preceding sentence (CAM-Chem CTM) along with a reference (Lamarque et al. 2012). Perhaps the reviewer is pointing out that we forgot to mention that the retrieval also requires a priori information for surface temperature and emissivity? We have now mentioned this in the text.

P5L133: Watch your usage of "layers" versus "levels" here and throughout. Retrievals are on layers, but are reported on levels for MOPITT. Note there are only 8 layers from 900 to 100 hPa (when surface pressure is greater than 900 hPa). The uppermost layer is 100 to 50 hPa.

We have modified the text so as to only refer to 'levels' (or 'levels of the profile', where elaboration is necessary for clarity) throughout, when talking about MOPITT profile levels. Thank you for pointing out the

missed details about the number of layers and the 50 hPa cap to the uppermost layer – we have clarified this in the text.

P5L141: Equation 1 is missing the error term.

We have used the form of the equation that is used in e.g. Deeter et al. 2017 (we have now referenced this accordingly when introducing Equation 1). We already outline in the preceding sentences that "The AK matrix…depends on the radiance weighting functions, instrument error covariance matrix, and a priori covariance matrix", and feel that introducing the error term here would add further complication to an already quite dense paper.

P6L163: "Is generally advised against" – please provide a reference.

The statement is made in the V6 data quality summary[1], in the "Data Filtering" paragraph on Page 2. However, since the data quality summary is not peer-reviewed, and no reference is given therein to support the statement, we have decided to remove the "is generally advised against" statement and stick to what is factual, i.e. that averaging together retrievals with different sensitivity profiles *will* dilute information coming from MOPITT radiances with information coming from the a priori. This then leads on to the next section about data filtering guidelines.

> The averaging together of retrievals with significantly different sensitivity profiles – as could be the case when averaging retrievals over land and water –  serves to dilute the information coming from the MOPITT observed radiances with information coming from the a priori, thus

P6L165: These supposed guidelines need to be clarified. While such restrictions may increase the average information content of individual soundings, the restrictions may decrease the information content of the system as a whole.

This is a fair and often overlooked point. We have included a statement at the end of the paragraph in Section 2.1.2 that outlines these guidelines to point this out (see screenshot in response to your comment G5 above). We feel that saying any more is beyond the scope of this study, since it is a trade-off that obviously depends on the purposes for which the data are being used. This point has been discussed further in response to the G5 comment above.

P6L173: Maybe this many significant figures are what are reported in the census, but it seems like too many. What if someone moves to Halifax?
* * *
[1] https://eosweb.larc.nasa.gov/sites/default/files/project/mopitt/quality_summaries/mopitt_level3_ver6.pdf

We have re-stated this as "with a population in excess of 315,000…":

> longitude = -63.58°3, latitude = 44.65°1). Situated on the Atlantic coastline, Halifax is the major economic center in Atlantic Canada, with a population in excess of 315,00016,701 in the urban core of Halifax Harbour

P6L174: Briefly describe the pollutants.

We have included a list of the trace gases monitored in the referenced study.

> (from 2016 census statistics). The pollution environment of Halifax, which is an intermediate port city, was characterized in detail by Wiacek et al., (2018 – trace gases of focus include $SO_x$, $CO_2$, $CO$, $NO_x$, $O_3$, $HC$, and $PM$); briefly, it showed no exceedances of regulated gaseous contaminants, but nevertheless a substantial contribution of shipping emissions that is comparable to or greater than emissions from the city's vehicle fleet and a nearby 500 MW power plant. All available MOPITT V7 L2 and L3 TIR-NIR files ("MOP02J"

P6L177: It appears that 2 more years of MOPITT V7 data are now available.

This was not the case at the time the bulk of the research was undertaken, and we feel that the length of the data record included in the study is enough to support the conclusions made.

P7L205: A brief explanation about the 4 MOPITT pixels is needed here.

This has been done.

> to the L3 product that they create, we filter these based on pixel number (each pixel corresponds to one of MOPITT's four along-track detectors) and channel-average signal-to-noise ratio (SNR), as is done at the V7 L3 processing stage to improve L3 information content by excluding observations from specific detector elements on MOPITT's detector array that were found to exhibit greater retrieval noise than the other

P8L223: I dislike the use of "true" for model values. Please modify throughout.

We have replaced all instances of $X_{true,sim}$ with $X_{tr,sim}$. We experimented with using different words altogether (i.e. $X_{model,sim}$) but found it to be less logically consistent with Eq 1 (when the 'real' true profile is referred to and to which we compare $X_{tr,sim}$ in the discussion of Section 3.1.3), and also the use of 'model' created confusion with what are actually the simulated profiles (i.e. $X_{sim}$)! We hope that this compromise is agreeable.

P13: Watch significant figures throughout.

We have taken care to limit values presented to a relevant/sensible number of significant figures.

Table 3: Is the purpose of p-values here to show that CO levels are changing in the MOPITT record?

The p-values cited in this table quantify the probability that the trend is zero (i.e. whether the trend can be considered statistically significant or not), as outlined in the table's caption.

**Table 3** Results from WLS regression analysis of seasonal mean L3W, L3L, L3O, L3O(water) and L3O(mixed) timeseries  for selected profile levels in DJF and JJA. Trend corresponds to the gradient of the WLS best-fit line; SE = standard error of trend ;  P-value = probability that the trend is zero; % change y⁻¹= mean percentage change in retrieved CO per year, calculated from WLS regression model

Briefly describe why OLS was used.

In this section we now use WLS (on the recommendation from a different reviewer) since it is less susceptible to outliers. We start the section by saying "to identify and compare temporal trends…" We are unsure what further description the reviewer would like?

Figure 6: "Next page" (?)

Thanks for spotting this – it is erroneous and has been removed.

Figure 8: Are the bounding boxes shown correct for a 1 degree box?

Correct – now explained in the caption to Figure 8. Thanks for pointing this out.

Ashpole and Wiacek analyze a MOPITT CO Level 3 (L3) pixel over one location – Halifax, Canada – and compare it to the Level 2 (L2) retrievals within the same 1 degree pixel. They use this coastal location to highlight instrument sensitivity differences between water and land that impact near-surface and profile analysis with the joint TIR- NIR product. The influence of different surface-types (land or water) on CO retrievals and the resulting trends is investigated. The authors find that sensitivity differences account for retrieval differences in JJA, but a CO gradient is likely the reason for differences in DJF. While the MOPITT team already provide recommendations to maximize information content for a studied region, Ashpole and Wiacek demonstrate the practical implications of retrieval differences. The study suggests L2 profile data over land is more appropriate than L3 profile for the small region around the coastal city of Halifax, particularly for the lower troposphere. The authors present a valuable supplementary guide for users of the MOPITT product.

Overall the analysis shows attention to detail and generally good statistical practices. The manuscript is well written and language is clear with very few technical corrections. I have some recommendations for reducing verbosity. Additionally, I recommend the following major comments be addressed before the manuscript is considered for publication.

**Major comments:**

1. Section 3.2: In my opinion, the most logically inconsistent part of the manuscript is the comparison of all L3O pixels with L3L and L3W . The surface pixel type (land, water or mixed) is given in the MOPITT level 3 product, and users should use information to filter data over such locations. I think the strength of this analysis would be to show that the improvement in information content if you use the L2 land data over coastal locations. Also - is there a difference between using L2 water and the L3 water retrievals, i.e. the extra days you gain from the "mixed" L3 pixel when creating L3W. The authors need to clearly justify why they are comparing with the combination of L3 pixel-types, or alternatively compare against the water-only pixels from Level 3.

We thank the reviewer for raising this, and apologise for not considering it in the original submission. We have completely re-written Section 3.2, and feel that the analysis benefits from now considering the subsets of L3 data separately. Our main findings are unaffected (if anything they are strengthened): even the days classified as 'mixed' in the original L3 product (which we refer to in the text as $L3O_{(mixed)}$), which represents the best option in the case of surface level retrievals in JJA owing to the lack of days where the original L3 product is classified as 'land', are characterised by markedly lower variability in retrieved surface level VMRs than in the L2 land data (which we call L3L), and the temporal trend identified remains a significant underestimation. We also evaluate the difference between using the L2 water data (which we call L3W), and the L3 days when the surface index is classified as 'water' ($L3O_{(water)}$), pointing out that, yes, temporal coverage is greater in L3W due to data effectively being reclaimed from days when the L3 surface index is 'mixed', but that this appears to make little significant difference when it comes to temporal trend analysis.

2. Trend analysis: The MOPITT profile layers are known to have drift (Deeter et al., 2017, doi:10.5194/amt-2017-71). Surface drift is about -0.7% per year, 800 hPa is -1% per year. The UT levels have positive drift - both the 400 and 200 hPa levels drift at greater than 1% per year. Drift should be corrected before trend analysis is performed. I did not see this mentioned in the manuscript.

We now take care to mention the drift in profile layers in Section 3.2.2 (see screenshot below). We do not correct our data for drift, which we feel is justified since we are simply comparing trends identified in the same dataset, as opposed to doing a detailed study about changing CO concentrations over the city of Halifax *per se*, and this is outlined in our discussion. We also include the drift values from Deeter et al. 2017 in our Table 3, for reference – most of the trends we identify appear to exceed the drift, at least for the timeseries that has the greatest sensitivity to the true profile (i.e. either L3L or L3W).

Material 8. It is important to note that MOPITT profile measurements are known to have a drift (Deeter et al., 2017), and this should be corrected for in the data if the focus of analysis is to use them to quantify temporal changes in CO over time. Since the intention of the WLS trend analysis presented here is more illustrative, namely to demonstrate trend differences in the data, we have not corrected for this drift. The results should therefore not be taken out of this context (as well as bias correction, verification against a range of other datasets would be required, especially given the large proportion of missing data). We do however provide the reported drift values in Table 3 for context, which shows that the majority of the trends that we have identified appear to be stronger than the measurement drift (at least for the dataset that has greatest retrieval sensitivity at the respective level of the profile). As noted in Section 2.1.1, the measurement drift

has been significantly reduced in the latest version of the MOPITT products to be released (Version 8; Deeter et al., 2019).

3. Ordinary Least Squares analysis is highly susceptible to outliers and end points. End points may in particular be impacting the slope calculations for L3L data in JJA (Figure 10, bottom panel). Instead, I recommend that the trend analysis be performed with weighted least squares (WLS), weighted by the standard deviation, which is much less susceptible to outliers. Additionally, the seasons that only have one day per 3 months can be de-weighted by using sufficiently large standard deviation.

Thank you very much for this suggestion. We have replaced our OLS results with results from WLS analysis. Our main findings are unaffected.

4. Please mention somewhere (for example on P6, L189) why the analysis is statistically valid, even though only 7% of the potential DJF days and 17% of the potential JJA days have measured CO.

This is a fair point. The high proportion of missing data, especially in DJF, is frustrating, but also unavoidable when using these data. It certainly makes it difficult to draw conclusions about the time period covered *as a whole*, but it does not preclude the making of conclusions about the temporal subset of data being studied – especially where differences are found to be statistically significant. We now make sure to point out that due to the high proportion of missing data, our conclusions should not be taken as being representative of the time period as a whole. We do this in the Data and Methods section (2.1.3 – screenshot below) and also re-iterate it in Section 3.2.2 when discussing the temporal trends we identify (see screenshot in response to your comment #2).

that were made during daytime hours. This yields a timeseries with one observation per day, when retrieval data were available within this gridbox. There are no retrievals available on 91 % of all days in DJF and 83 % of all days in JJA for the period covered. This is a result of both 1) MOPITT's polar orbit limiting temporal resolution to ~3 days over most of the globe; and 2) on days when the satellite's swath does encompass Halifax, retrievals either not being made due to cloud coverage, or discarded due to data quality issues. While this does not prevent a meaningful comparison of available retrievals, it does mean that caution is needed when using them to draw conclusions about the time period covered as a whole, which is something that we do not attempt to do. (there are no retrievals available on 93 % of all days in DJF and 83 % of all days in JJA

5. I am uncomfortable with the authors extending this analysis to other regions (e.g. P2, L35; P20, L605 to L607). Before extending these results to other coastal locations, other types of locations need to be analyzed (tropical, temperate). Please add a comment that recommends similar analysis be performed for other coastal sites before using MOPITT surface CO.

We apologise for making these claims, and are appreciative of the suggested alternative wording. There are statements along these lines on 3 separate occasions in the original submission (in the abstract, introduction, and conclusion), and we have toned the wording down in all cases:

Abstract:

the findings also apply to MAM and SON. Although we focus only on the city of Halifax, our results imply potentially large differences in the results of near-surface CO analysis using the L2 and L3 datasets for other cities that are situated within a coastal L3 gridbox, among which are some of the most populous in the world. The results that we report here suggest that similar analyses be performed for other coastal cities before using MOPITT surface CO.

Introduction:

> likely to be targets for analyses of temporal trends in air quality indicators. The results that we report here are therefore relevant to each of these cities.

Conclusion:

> in order to maximize the information content of L3 data in coastal gridboxes. Although our study has only focused on the city of Halifax, the results suggest that similar studies be performed for other coastal L3 gridboxes before using MOPITT surface CO,  since these contain 6 of the top 10 and 43 of the top 100 agglomerations by population and are therefore likely targets for analysis of temporal changes in air pollution indicators such as CO, especially near to the surface. The

6. There was a change in MOPITT processing after the cooler failure in May 2001. Although a homogenized record is attempted, there remains a small step-change in data. I suggest to use data only from the latter part of the record starting August 2001, especially for the trend analysis where step changes could have large impact.

Thanks for pointing this out. We have re-done all analysis, only using data after August 2001 as recommended. The main results are unaffected by this change, although there are small differences in the specific details. We outline the reason for data truncation in our Data and Methods section:

> fleet and a nearby 500 MW power plant. All available MOPITT V7 L2 and L3 TIR-NIR files ("MOP02J" and "MOP03J" files, respectively) were downloaded from the NASA Earthdata portal (https://search.earthdata.nasa.gov). There is a small inconsistency in the data record before and after an instrumental reconfiguration in 2001 (Drummond et al., 2010); we therefore discard all data prior to this reconfiguration. The remaining data covers the period 2000-08-25 to 2017-03-05. At the time of

7. The authors have missed an opportunity to quantify the improvement in retrievals due to the inclusions of NIR between L3L and L3W . In the JJA comparisons, when the CO gradients and a priori suggest there are no expected differences in the CO.

In response to a comment from another reviewer, we have now conducted additional analysis using the TIR-only dataset, to demonstrate that the land-water contrast is present in that dataset and is therefore not purely a result of additional information gained from the NIR being limited to retrievals over land in the joint TIR-NIR product. This is outlined in Supplementary Material 1. However, we do not feel that we can go as far as quantifying the improvement in retrievals due to the inclusion of NIR between L3L and L3W without significant extra analysis which will distract from the main point of the paper.

We do explicitly address the role of NIR in Section 3.1.2. (Climatology of land-water retrieval sensitivity differences), however. We outline that LT sensitivity enhancements due to the inclusion of NIR is limited to

retrievals over land and that this therefore likely has an impact on the LT land-water sensitivity contrast. However, our supplementary analysis with TIR-only data (as shown in Supplementary Material 1) shows that the LT land-water difference in AKs is comparable to what it is in the joint TIR-NIR product used in our study, highlighting that thermal contrast differences must be important.

> for   L3W . Since our analysis is conducted using the joint TIR-NIR product, it is important to bear in mind that the benefit of enhanced LT sensitivity due to the incorporation of NIR is limited to retrievals over land, so this will also have an impact on the AK differences presented above. However, a land-water retrieval sensitivity contrast of comparable magnitude to that presented here is also evident in the TIR-only product, reinforcing the primary role of thermal contrast differences (see Supplementary Material 1).

**8.** Figure 6 and associated discussion. I suggest to split the logXtrue - logXap into different average profiles for land and water because Figure 7 suggests that in JJA both difference would be negative while for DJF, the land difference would be negative but the water would be positive. This would help support the overall argument that DJF sees real CO gradients, while JJA is impacted by sensitivity differences.

Thank you for the suggestion, but in this instance however, we have chosen not to modify the analysis. Our justification is as follows: the analysis of simulated profiles presented in Figure 5 and discussed in Section 3.1.3 is intended only to demonstrate how averaging kernel differences can affect retrieved VMRs. Thus, AKs are the only variable between simulated retrievals for land and water. We feel that by introducing further variables, in the form of different Xtrue and Xapr profiles, the main point about AK control would be lost. Admittedly, this could be an additional analysis component (i.e. a second set of simulated profiles for analysis – perhaps with AKs held constant over land and water?), but we do wonder whether it would distract from the main thread of the paper and whether this is really necessary, especially given comments from a different reviewer about the length of the paper. This is obviously something that we can be flexible about however, if the reviewer feels strongly that this is an angle we should include.

**Minor comments:**

P1, L9: Add latitude and longitude to location in abstract.

This has been done:

> We compare MOPITT Version 7 (V7) Level 2 (L2) & Level 3 (L3) carbon monoxide (CO) products for the $1^o$ x $1^o$ L3 gridbox containing the coastal city of Halifax, Canada (longitude = $-63.58^o$, latitude = $44.65^o$),

P1, L18 L30 and elsewhere in manuscript: Change "surface profile level" to just "surface level" - otherwise it should be "surface level of the profile".

Thanks for the suggestion – we have changed most instances in the manuscript to "surface level", apart from a few which we have changed to "surface level of the profile" where the elaboration is useful.

P2, L41: Remove "primary" from "The primary target" because MOPITT only retrieves CO.

This has been done:

> the composition of the Earth's atmosphere from space. The  target gas for MOPITT is carbon monoxide (CO), whichemitted from a range of anthropogenic (e.g. fossil fuel use) and natural (e.g.

P2, L46: Add in a sentence about the secondary production of CO from VOCs.

This has been done:

> monoxide (CO), whichemitted from a range of anthropogenic (e.g. fossil fuel use) and natural (e.g. wildfires) sources, produced via the oxidation of methane and other volatile organic compounds, and has an atmospheric lifetime of weeks to months depending on season and location (e.g. Duncan et al., 2007).

P2, L48: "...since launch in December, 1999".

This change has been made:

> et al. (2013) for a comparison of CO trends from four satellite instruments), the unique strength of MOPITT lies in its nearly unbroken record of observations since launch in December 1999. This makes MOPITT data

P4, L114 to 116: It is important to note here that the improved sensitivity occurs only over land for the joint TIR-NIR product. Also, largest difference would consequently be expected between land and water with the joint product, so the study presented in this manuscript is expected to show an upper bound on the differences in retrievals between land-types.

Thanks for pointing this out. We now outline this in the text:

lower troposphere (LT), to which TIR radiances are typically less sensitive (Pan et al., 1995, 1998). NIR radiances can, however, only be exploited in daytime scenes over land. Our results are based on analysis of the TIR-NIR combined product, owing to its greater sensitivity to LT CO compared to the TIR- and NIR-only products which are also available (e.g. Deeter et al., 2017). Owing to the increased LT sensitivity from NIR radiances being limited to retrievals over land, we expect that the results presented here show an upper bound on the retrieval differences between surface types within our coastal L3 gridbox of focus, and the consequent effects on sample statistics and temporal trends that we outline. Differences are still found in the

TIR-only product, however, and we outline these in Supplementary Material 1. We restrict our analysis to

P4, L127: The simulation climatology resolution is 1.9◦ × 2.5◦, not 1◦ × 1◦.

Thanks for this clarification – the change has been made:

are derived from a monthly CO climatology for the years 2000–2009, simulated with the Community Atmosphere Model with Chemistry (CAM-chem) chemical transport model at a spatial resolution of 1.9º x 2.5+º (Lamarque et al., 2012) and then spatially and temporally interpolated to the time and location of the

P6, L172: Add degree symbols to the latitude and longitude locations.

This has been done:

Our analysis is based on MOPITT retrievals over the city of Halifax in Nova Scotia, Canada (Figure 1: longitude = -63.58º3, latitude = 44.65º1). Situated on the Atlantic coastline, Halifax is the major economic

P9, L274: I think it should be "MT and UT".

Thank you for spotting this! The change has been made:

retrievals obtained at the same time within the 1º x 1º L3 gridbox containing Halifax. Mean ΔRET values are closer to zero and generally less significant in the MT and ULT (represented by the 600 hPa and 300 hPa

P10, L299: I was confused by "the widest part of each AK" - Did you mean in the X-direction or the Y-direction? Please clarify.

Apologies for the confusion – we have modified this part of the AK description to make it clearer:

> profiles in L3L and L3W analysed in the previous section. Each curve corresponds to a row of the AK matrix and represents the sensitivity of the corresponding level of the retrieved profile to each level of the true CO profile, with  the widest part of each AK in the x-direction (when a peak is evident) indicating the portion of the true profile that the corresponding level of the retrieved profile is most sensitive to. The sum of the elements in each AK row  represents the overall sensitivity of the retrieved

P11, L328: Do you mean differences in the MT?

We apologise that this sentence is unclear. We did mean to refer to the LT here, and were implicitly referencing the fact that the 300hPa AK reaches negative values at the surface. We have reworded this to make the point explicitly and (hopefully) improve clarity:

> level, which is around 3 times lower than that over land. As in DJF, UT AKs are quite similar, except for relatively small differences at the surface, where the 300 hPa AK over land actually indicates negative sensitivity. Differences in mean DFS values for retrievals in L3L and L3W are greater in JJA (-

P12, L346 to L347: Reword to be a little clearer, e.g. "The Tskin difference approaching 20° K between JJA L3L and DJF L3W likely..."

We have reworded this for clarity:

> inverted temperature profile. The T$_{skin}$ increase approaching 20° K between DJF and JJA likely also accounts for the relatively greater overall true profile sensitivity (indicated by DFS values) in JJA for L3L than in DJF for  L3W . Since our analysis is conducted using the joint TIR-NIR product, it is important

P12, L370: Reword to clarify, e.g. "Available CAMSRA data covers 2003-2016, so a subset of the available MOPITT data (2000-2017) are considered in simulation experiments."

We have reworded this for clarity:

> retrievals are solely a result of differences in AKs.  Because available CAMSRA data only cover the years 2003-2016,  only a  subset of the retrieval pairings considered in earlier sections (which span the period 2001-2017) are  simulated.

P13, L377: Remove the "respectively" explanation in brackets because there is only one number.

This has been done.

P13, L378: "...sensitivity to Xtrue,sim in L3L than L3W :..." P15, L463: Change all-caps TRUE.

We have renamed $X_{true,sim}$ to $X_{tr,sim}$ on recommendation of another reviewer, so this no longer seems relevant?

P16, L472 to L473: This is the first time the data is separated in this way (L3L>L3W and L3W >L3L). Mention why this is necessary. Why is it not separated this way in the other sections - for example Figure 2 shows a mean response.

We have included justification for this data separation in the text (see screenshot below). To us, this seemed like the logical way to answer the question of whether circulation differences could be generating the CO gradients that MOPITT is detecting. In the other sections it is the mean of the L3L and L3W datasets that is being compared, and shows clear differences.

> We compare composite mean wind patterns across Nova Scotia using ERA-Interim data for days when retrieved surface level VMRs in L3W are greater than in L3L (L3W > L3L) and days when they are less (L3W < L3L), since a clear shift in wind direction on these days would support the case that atmospheric transport plays a role in generating differences in retrieved CO amounts over land and water.  These are

P16, L491 to L496: This description needs a little clarification. Could it be explained that the density of days where L3W <L3L is higher at the beginning of the record, while the density of days where L3W >L3L is higher at the end of the record for JJA. In DJF, the density of L3W >L3L is consistent across the record.

Thank you for pointing out the lack of clarity here. On closer inspection, what we had previously written was not entirely accurate, so we have reworded accordingly, with your suggestion in mind. Our main point still holds however, that days when L3W < L3L are concentrated at the start of the timeseries in JJA.

While there is no obvious circulation difference at the surface in JJA between days when L3W > L3L and L3W < L3L, there is a difference in  the distribution of these days throughout the analysed MOPITT timeseries, which spans 16 JJA seasons from 2001 to 2016. 16 of the 19 days when L3W < L3L occur in the first half of the timeseries (i.e. before 2009), whereas days when L3W > L3L are spread  more evenly throughout (33 of the 65 days (51%) occur before 2009). In DJF there is no such difference, with roughly 50% of days occurring before and after 2009 in each case. This is something we explore further in Section 3.2.2.~~The same is not true in DJF (mean retrieval years = 2008 and 2007 respectively). As we show in Section 3.2.2, this appears to be a function of true LT CO concentrations over Halifax decreasing across the time period studied, which is detectable in JJA by the retrievals in L3L owing to their greater near-surface sensitivity, but not by the retrievals in L3W, which are tied to an a priori value that is too low at the start of the timeseries and too high at the end.~~

P17, Section 3.2.1: Discuss/show p-values for all comparisons between means.

We have rewritten this section entirely, and mention where the mean differences are statistically significant.

P18, L549: Consider discussing as "gradients for L3W " etc. rather than introducing more acronyms. This would also help when reading Table 3 as the new acronyms are not used there.

We have rewritten this section entirely, and remove the acronyms, instead discussing "trends in L3W" etc.

P20, L604: "... before analysis of profile values in order to...".

This change has been made:

**4. Conclusions.**

Users of MOPITT products are advised to filter the data before analysis of profile values in order to maximize the influence of satellite measurements and minimize the impact of a priori CO concentrations on results (MOPITT Algorithm Development Team, 2017; Deeter et al., 2015). In particular, it is advised that retrievals

P20, L605: "In particular, when investigating coastal cities, it is advised that retrievals over land..."

In the line that the reviewer refers to here, we mention the previously-advised filtering criteria from the MOPITT Algorithm Development Team (2017) and Deeter et al. (2015) (references to these have been added). Coastal cities are not mentioned in this advice. So the suggestion is invalid in this case.

Table 3: Add units for m and SE.

This has been done.

Figure 4: Add error bars for standard deviation.

This has been done.

Figure 8: "Yearmean" is an unusual metric. I suggest "Center year".

We have removed this from the figure and no longer discuss Yearmean, instead quantifying the number of days for each sample in the first and second halves of the timeseries.

Figure 9: What are the error bars?

(Note that this is now Figure 8) These are boxplots, so the errorbars correspond to the range of values covered between the minimum value (excluding outliers) and q1; and between q3 and the maximum value (excluding outliers). We have stated in the caption that these are boxplots to avoid confusion.

**Suggestions to reduce verbosity:**
Thank you for these!

P5, L155: Remove end of sentence from "in the case of" onwards - redundant.

This change has been made.

> each L2 retrieval is tagged according to whether it was performed over land, water, or a combination of the two ("mixed") . The surface index of each L3 gridbox is then based on the L2 retrievals that fall within the relevant

P8, L227-L228: Remove "For simulated..." onwards - redundant.

This change has been made.

> the mean of the a priori fields that correspond to the temporally coincident L3L and L3W pairings ; and A is the retrieval averaging kernel from L3L or L3W.  Thus, any differences between each pair of simulated retrievals ($X_{sim,L3-L}$ and $X_{sim,L3-W}$) are solely a result of differences in A, since $X_{true,sim}$ and $X_{apr,sim}$ are identical for both. Simulations are initially performed on $\log_{10}$(VMR) for consistency with the

P12, L348 to L361: This is a repeat of earlier and later information. Consider whether any or all of it is really needed here.

We have removed a lot of the speculation and forward referencing from this section, but keep some of the summarising as we feel it is useful to pull the key results together at this stage before digging deeper.

> That  retrieval sensitivity contrast between L3L and L3W is most pronounced in the LT is consistent with the finding that retrieved CO profiles in L3L and L3W show the greatest  differences in the LT ~~(Section 3.1.1, Figure 2). In DJF, the retrieval is more sensitive to the true profile in the LT in L3$_W$ (Figure 3, top row) and retrieved LT VMRs are greater in L3$_W$ than L3$_L$. In JJA, the retrieval is more sensitive to the true profile in the LT in L3$_L$ (Figure 3, bottom row), and retrieved LT VMRs are less in L3$_L$ than L3$_W$. This implies that in DJF, LT CO retrievals in L3$_L$ are weighted more heavily towards an a priori profile in which LT VMRs are lower than in the true CO profile; while in JJA, LT retrievals in L3$_W$ are weighted more heavily towards an a priori profile in which VMRs are greater than in the true profile. We explore this point in detail in Section 3.2.2.(see Supplementary Material 1)~~. Likely causes for this could be day-to-day changes in atmospheric conditions (i.e. temperature or water vapour profiles), or random instrumental or retrieval noise.

P12, L366 to L369: Remove sentence starting "For each L3L-L3W ..." because repeats information from the methods section and is not needed here. Can just start sentence on line 369 with "Recall that any differences..."

This change has been made.

> outlined in Section 2.2.  Recall that any differences between each pair of simulated retrievals are solely a result of differences in AKs. Because available CAMSRA data only cover

P13, L382 to L383: Ending of sentence starting "as would be expected from..." can be removed because it is a repeat of L378 to L380.

This change has been made.

> much closer to $X_{true,sim,sfc}$ (32.21 ppbv higher vs 57.13 ppbv higher). Both  $X_{sim,L3Lsfc}$ and $X_{sim,L3W}$ values indicate that $X_{true,sim,sfc}$ is lower than $X_{apr,sim,sfc}$ at the surface, but $X_{sim,L3-L,sfc}$ gives the closer estimate, as would be expected . In both cases, a portion of the

P13, L384 to L391: Unsure that this adds any important points to the discussion, consider removing or shortening.

We have removed a lot of the explanation here as we agree that it does not add anything important to the discussion – we simply mention the interlevel correlation of the retrieval, which it is important to be aware of.

> would be expected . In both cases, a portion of the overall $X_{sim,sfc}$ departure from $X_{apr,sim,sfc}$ at the surface level (in other words, "value added" over the a priori) originates at other  levels of the profile, and not the surface level itself. This is a result of the surface level AK$_{sfc}$ being nonzero at other  levels, and is a function of the inter-level correlation of the original retrieval, which is linked to the a priori covariance matrix used in the retrieval (Deeter et al., 2010). ~~Because AK$_{sfc}$ has a well-defined peak at the surface L3$_L$ (Figure 5), while AK$_{sfc}$ over L3$_W$ actually indicates greatest retrieval sensitivity (albeit very weak) to a broad region from 900 – 400 hPa, the surface accounts for a greater proportion of the $X_{sim,L3-L,sfc}$ departure from $X_{apr,sim,sfc}$ (55 % vs 25 % – see Appendix B for full derivation of these values) while the $X_{sim,L3-W,sfc}$ departure from $X_{apr,sim,sfc}$ has a greater contribution from the lower-to-mid troposphere as a whole.~~

P16, L496: "As we show in Section 3.2.2,..." - Is this really needed here, or could you leave it till Section 3.2.2.

We have removed the lengthy explanation and now just state that "this is something we explore further in Section 3.2.2".

retrieval year = 2008). In DJF there is no such difference, with roughly 50% of days occurring before and after 2009 in each case. This is something we explore further in Section 3.2.2. The same is not true in DJF (mean retrieval years = 2008 and 2007 respectively). As we show in Section 3.2.2, this appears to be a function of true LT CO concentrations over Halifax decreasing across the time period studied, which is detectable in JJA by the retrievals in L3$_L$ owing to their greater near-surface sensitivity, but not by the retrievals in L3$_W$, which are tied to an a priori value that is too low at the start of the timeseries and too high at the end.

P17, L510 to L511: Remove end of sentence beginning "primarily because..." - repeat of information in the beginning of the sentence.

Section 3.2 (which this comment refers to) has been entirely re-written.

P17, L511 to L514: Remove "Thus, there is..." - it is unnecessary.

Section 3.2 (which this comment refers to) has been entirely re-written.

P17, L524 to L526: Consider removing "(mean surface level..." to "...248 for L3O)".

Section 3.2 (which this comment refers to) has been entirely re-written.

P19, L581 to L587: There is no significant difference between trends at 300 hPa , so there is no need to explain why they might be different. This paragraph just needs to mention the main point that trends at 300 hPa are positive.

Section 3.2 (which this comment refers to) has been entirely re-written, and we take care to limit our discussion to significant results.

Table 2: AKsfc are in Figure 5 so can be removed. Consider adding other measures from Table 2 to Figure 5 and removing Table 2.

We have removed Figure 5 since the AK values are in Table 2. We choose to keep Table 2 as opposed to Figure 5 as it seems like the better way to present the data in this case.

---

## Author Response (AR2)

We are very thankful to anonymous reviewer #2 for their second review of this paper and pointing out the need for some technical corrections. We have made all of these corrections in the resubmission, and outline the corrections below.

Reviewer comments are in black.

Our responses are in blue.

We provide a screenshot of the track-changes document to show the changes that we have made (in outlined boxes). In these, text that is removed is struck through and coloured red, while new text is underlined and coloured blue. (Note that the line numbers are different in our screenshots to those referenced by the reviewer, because the line numbers the reviewer references are from the tracked changes version of the revised manuscript whereas the line numbers in the screenshot are from the revised manuscript file, which had the tracked changes resolved.)

Ian Ashpole (on behalf of both authors).

Received: 13 May 2020

Technical corrections:
Lines correspond to the tracked changes version of the updated manuscript.

Line 148: "Where surface pressure is below 900 hPa, only 8 profile levels are retrieved" Change to: "Where surface pressure is below 900 hPa, less than 10 profile levels are retrieved" because it depends on what that surface pressure is, e.g. <900, <800 etc.

Thanks for pointing this out – an obvious oversight on our behalf! We have made the appropriate change.

> 100 hPa (the uppermost level covers the atmospheric layer from 100 to 50 hPa) and a floating surface pressure level. Where the surface pressure is below 900 hPa, less than 10 profile levels are retrieved. Reported values represent the mean CO volume mixing ratio (VMR) in the layer immediately above that level. Retrievals are initially performed on a $\log_{10}$(VMR) scale, owing to large CO
> 145    variability in the atmosphere.

Line 458: Should it be "smaller magnitude in DJF *than* JJA"?

Correct – thanks for spotting this. We have made the change.

> 415    at least in the LT. However, while $\Delta$SIM values are of a much smaller magnitude in DJF than in JJA, $\Delta$RET is actually of a similar magnitude in both seasons. The question therefore arises as to why $\Delta$SIM is so different from $\Delta$RET in DJF, and why it is of much smaller magnitude in DJF  than JJA. Considering the terms of

[revised manuscript text omitted]